# A Controllable Examination for Long-Context Language Models

**Yijun Yang**[*1,2,6], **Zeyu Huang**[*1], **Wenhao Zhu**[3], **Zihan Qiu**[4],
**Fei Yuan**[2], **Jeff Z.Pan**[1], **Ivan Titov**[1,5]

[1]University of Edinburgh [2]Shanghai Artificial Intelligence Laboratory [3]Nanjing University
[4]Qwen Team, Alibaba Group [5]University of Amsterdam [6]Aveni.ai

## Abstract

Existing frameworks for evaluating long-context language models (LCLM) can be broadly categorized into real-world applications (e.g, document summarization) and synthetic tasks (e.g, needle-in-a-haystack). Despite their utility, both approaches are accompanied by certain intrinsic limitations. Real-world tasks often involve complexity that makes interpretation challenging and suffer from data contamination, whereas synthetic tasks frequently lack meaningful coherence between the target information ("needle") and its surrounding context ("haystack"), undermining their validity as proxies for realistic applications. In response to these challenges, we posit that an ideal long-context evaluation framework should be characterized by three essential features: 1) seamless context: coherent contextual integration between target information and its surrounding context; 2) controllable setting: an extensible task setup that enables controlled studies—for example, incorporating additional required abilities such as numerical reasoning; and 3) sound evaluation: avoiding LLM-as-Judge and conduct exact-match to ensure deterministic and reproducible evaluation results. This study introduces **LongBioBench**, a benchmark that utilizes artificially generated biographies as a controlled environment for assessing LCLMs across dimensions of *understanding*, *reasoning*, and *trustworthiness*. Our experimental evaluation, which includes **18** LCLMs in total, demonstrates that most models still exhibit deficiencies in semantic understanding and elementary reasoning over retrieved results and are less trustworthy as context length increases. Our further analysis indicates some design choices employed by existing synthetic benchmarks, such as contextual non-coherence, numerical needles, and the absence of distractors, rendering them vulnerable to test the model's long-context capabilities. Moreover, we also reveal that long-context continual pretraining primarily adjusts RoPE embedding to accommodate extended context lengths, which in turn yields only marginal improvements in the model's true capabilities. To sum up, compared to previous synthetic benchmarks, Long-BioBench achieves a better trade-off between mirroring authentic language tasks and maintaining controllability, and is highly interpretable and configurable. [1]

## 1 Introduction

Long-context language models (LCLMs) have become increasingly important in recent years. They not only enhance pure text-based applications with lengthy documents [10, 11, 12, 13, 14] but also lay the vital groundwork for developing stronger multimodal language models that can integrate and

---

[*]Equal contribution.

[1]The code and data are available at `https://github.com/Thomasyyj/LongBio-Benchmark`.

Table 1: Comparison of long-context Benchmarks. Here cheap means that the benchmark is cheap to construct. *: We only consider the non-synthetic task within the benchmark.

| Benchmark | Cheap | Seamlessness | | Controllability | | Soundness | |
| | | Fluent | Coherent | Configurable | Extendable | Leakage Prev. | Reliable Metric |
|---|---|---|---|---|---|---|---|
| L-Eval [1] | ✗ | ✓ | ✓ | ✗ | ✗ | ✗ | ✗ |
| LongBench-v2 [2] | ✗ | ✓ | ✓ | ✗ | ✗ | ✗ | ✓ |
| NoCha [3] | ✗ | ✓ | ✓ | ✗ | ✗ | ✗ | ✓ |
| ∞-Bench* | ✗ | ✓ | ✓ | ✗ | ✗ | ✗ | ✗ |
| Helmet* [4] | ✗ | ✓ | ✓ | ✗ | ✗ | ✗ | ✗ |
| BABILong [5] | ✓ | ✓ | ✗ | ✗ | ✓ | ✓ | ✓ |
| RULER [6] | ✓ | ✗ | ✗ | ✓ | ✓ | ✓ | ✓ |
| Michelangelo [7] | ✓ | ✗ | ✗ | ✓ | ✓ | ✓ | ✓ |
| NoLiMa [8] | ✓ | ✓ | ✗ | ✓ | ✗ | ✓ | ✓ |
| MRCR(OpenAI) [9] | ✓ | ✓ | ✓ | ✓ | ✗ | ✓ | ✓ |
| LongBioBench | ✓ | ✓ | ✓ | ✓ | ✓ | ✓ | ✓ |

process diverse data types [15, 16, 17]. Nevertheless, evaluating long-context capacity remains a challenging problem that hinders consistent progress in the field.

Existing long-context evaluation benchmarks are primarily of two types: **natural** and **synthetic**. In terms of natural benchmarks, Karpinska et al. [3] collects different novels read by annotators and asks them to annotate true/false of the narrative phenomenon. Li et al. [18] collects the latest documents post year 2022 and crafts questions for them. Though they provide authentic language samples, they are expensive to collect and annotate and susceptible to data contamination. Moreover, their inherent complexity makes them hard to characterize for controlled studies. Thus, the model's performance on such benchmarks often fails to shed sufficient light on the model's underlying bottlenecks. In other words, while the data is genuine, it does little to pinpoint precisely how and why a model might struggle, offering limited insights into how its long-context capabilities can be improved. Moreover, since their questions are crafted by human annotators, the task is frozen after annotation and thus unextensible to new evaluation scenarios.

On the contrary, synthetic benchmarks are highly controllable. One can isolate specific aspects they want to test with carefully designed synthetic tasks. Benchmarks like RULER [6] allow practitioners to systematically adjust variables and examine their targeted hypotheses about the model's potential behaviors. However, existing synthetic benchmarks mostly follow the Needle-In-A-Haystack (NIAH) [19] format, where the context is usually **non-coherent** because the needle (the information to be recalled) is semantically unrelated to the continuous context. We show in Sec. 4.2 that this makes the benchmark less challenging when the tasks become more difficult, as it potentially presents shortcuts for the model to retrieve the target information. Meanwhile, the complexity of the needle itself is another limitation. As we will show in Sec. 4.1, the *numerical needle*, which is employed by NIAH [19] and RULER [6] benchmarks, is much easier to retrieve than other types of information. This implicit bias makes them worse proxies of real-world applications, hindering the development of stronger models.

These observations underscore a critical need for a more comprehensive evaluation framework that provides a better trade-off between the authenticity of natural data and the controllability of synthetic tasks. Building upon discussions above, we summarize three features for ideal LCLM evaluation benchmarks and show the comparison between benchmarks in Table 1: (1) **Seamless context**: Unlike most existing synthetic benchmarks, we argue that the target information (the needle) should be *seamlessly* embedded into the long context to prevent potential shortcuts that could hack the benchmark. That means the needle should be better in a fluent natural language, and the needle should be semantically coherent with the context haystack. (2) **Controllability** The benchmark should be *configurable* to enable controllable experiments and *extensible* to simulate

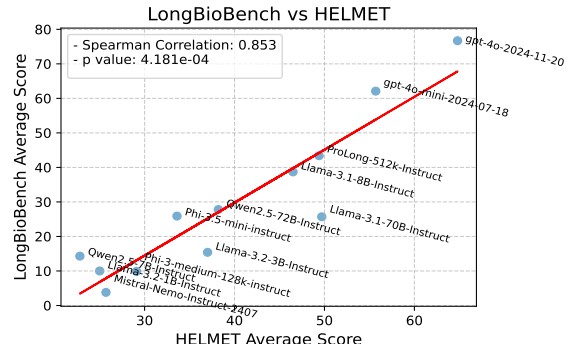

Figure 1: **Our synthetic LongBioBench has a high correlation with the real task HELMET [4].** The performance is tested on 128K context length from 12 overlapped models.

newly emerging tasks. (3) **Soundness**: The task should be free from parametric knowledge and generated on the fly to prevent data contamination. The metric should be accurate for sound evaluation instead of undeterministic evaluators such as LLM-as-Judge.

Inspired by Allen-Zhu and Li [20], this paper proposes **LongBioBench**, employing fictional biographies as a playground to examine existing long-context language models. In our framework, a single configurable biography (the "needle") is seamlessly embedded within a set of other configurable biographies (the "haystack"), enabling diverse and scalable task design grounded in each bio's rich factual content. Notably, we show in Fig. 1 that the evaluation scores on our purely synthetic benchmark exhibit a stronger correlation (0.853) with the scores of HELMET [4], which employs real-world tasks, than with other existing purely synthetic dataset RULER [6] (0.559). The task-specific correlations and the analysis are provided in App. G.1.

Furthermore, after evaluating 18 LCLMs, we reveal that (1) long context modeling faces some unique challenges. They usually struggle with numerical reasoning, constrained planning, or trustworthy generation, even though they are capable of retrieving relevant information (Sec. 3.3). (2) Using non-coherent context or numerical needles could prohibit the benchmark from revealing the true capability of LCLM, especially when tasks become more challenging (Sec. 4.2). (3) The density of distractors is another bottleneck for the performance of LCLMs (Sec. 4.4). (4) Finally, by testing our benchmark during the pretraining process, we show that the performance saturates at the early stage and challenge the current routine of continuing pretraining on many long context data (Sec. 4.3). In summary, though LongBioBench is not a perfect proxy for real-world tasks, we hope it can help the community deepen the understanding of LCLM behaviors to develop stronger models.

## 2 LongBioBench

### 2.1 Desiderata: What is an ideal benchmark for evaluating LCLM?

Before formally introducing LongBioBench, we first posit that an ideal synthetic benchmark for LCLM should satisfy the following three properties to address the flaws existing in most benchmarks.

(1) **Seamless context** The needle should be fluent in natural language and coherent within the context to prevent potential shortcuts that ease the task. Though most existing synthetic long-context benchmarks insert a needle into an irrelevant context (e.g RULER [6], BABILong [5]), we argue that such a construction method may destroy the harmony of the original context, thus causing some implicit shortcut for LCLMs to locate the needle and making the evaluation biased. Specifically, we show in Sec. 4.1 that LCLMs may be sensitive to retrieving numerical needles and in Sec. 4.2 that performance on incoherent needles is easier to retrieve compared with coherent needles as the difficulty of the tasks increases. Therefore, we propose that the inserted needle should be in fluent language, the same as the haystack, and should be logically coherent with the context. (2) **Controllable settings** The benchmark should be configurable to perform controllable ablation and extensible to simulate diverse tasks, allowing researchers to systematically investigate the internal dynamics of language models. Ideally, this extensibility should enable researchers to isolate and examine different prerequisite capabilities (e.g., arithmetic reasoning vs. retrieval skills). Despite the importance of these properties, we found that few existing synthetic benchmarks emphasize both configurability and extensibility. (3) **Sound evaluation** The evaluation should be unconfounded by parametric factual knowledge, and the metric should be reliable. To ensure reliable evaluation, we propose that the task should be free from reliance on the model's parametric factual knowledge to prevent contamination (e.g., a model may find it easier to identify an inserted needle if it has already memorized part of the haystack). In addition, the evaluation metric should be objective and reliable, avoiding using non-deterministic measures such as LLM-as-Judge and unexplainable metrics such as perplexity [21].

### 2.2 Data Construction

A data point for long-context evaluation could be composed of the following components: (1) a long context containing the needles (the information to answer the query) and the haystack (all other irrelevant information); (2) one or a few questions asking the model to understand, retrieve, or reason according to the needle in the context; and (3) the ground truth answer to the question. Overall, LongBioBench's context and needle are both artificial biographies. And the question in

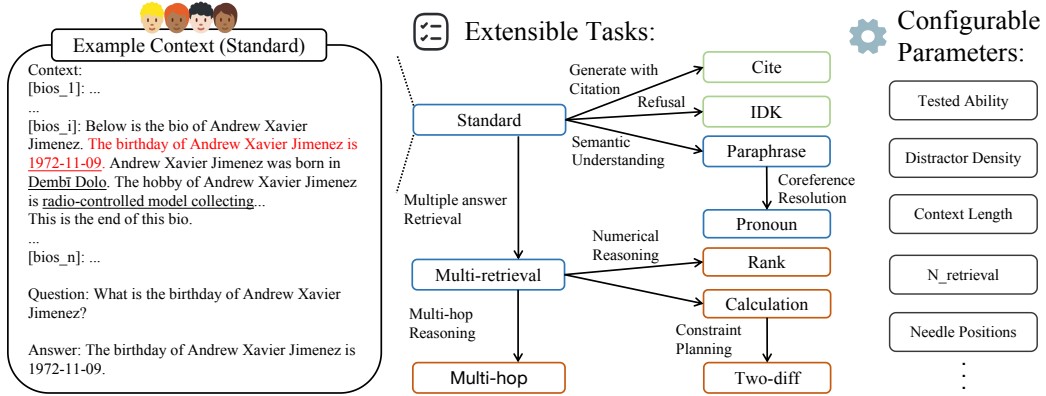

Figure 2: The example of our data (left), the supported configs and the extensible tasks (right). The underlined text shows the inserted attributes. The color of different tasks marks the category of the task.

LongBioBench could vary, from naive retrieval (e.g., retrieve specific information from the needle bio) to elementary numerical reasoning (e.g., find two people with a given age difference).

We use the simplest version, referred to as *Standard* version, as an example to show how a single data point is constructed. An illustration of our proposed context is shown on the left of Fig. 2, and the detailed construction progress is explained in App. B. Specifically, given a task, we first generate needle and haystack biographies using a biography generator. The needle biography is inserted into the haystack to form the context. The corresponding questions and answers are generated alongside the needle biographies, yielding a complete data point. For the biography generator, it typically samples six attributes from predefined attribute pools and fills them into human-written templates to produce coherent biographies for each individual. We also provide flexible configuration options at each step of the construction process to ensure controllability. For instance, we can adjust the distractor density when generating the haystack bios or define the number of required needles when constructing the needle bios. We report the statistics of the dataset in App. C.

The proposed benchmark closely matches the desiderata aforementioned. For **seamless context**, our generated biographies are ensured to be naturally fluent and maintain the coherence of the entire context. Regarding **controllability**, the benchmark is highly modular and configurable, allowing us to do isolated ablation studies to probe the model's behavior. The benchmark is also extensible, as the richly factual nature of the context enables a wide range of downstream tasks, depending on how the embedded knowledge is manipulated (further discussed in the next section). A challenging extended task: *In-context Learning* is introduced in Sec. 4.5 as an example. Finally, we ensure **sound evaluation** because the final answer in our benchmark can be easily verified through exact match. We also prevent contamination because all biographies are artificial and can be generated on-the-fly, thus avoiding reliance on any parametric knowledge memorized by the model.

## 2.3 Task Description

This subsection will explain how we extend to have all the tasks in LongBioBench. The tasks are split into three categories: understanding, reasoning, and trustworthiness, representing the core capabilities required to solve the tasks. An overview of how each task is extended is shown on the right of Fig. 2. The detailed description for each task is in App. D . Examples are illustrated in Table 2.

**Long Context Understanding**   One basic skill for long-context language models is understanding the user's query and retrieving relevant information from the long context. To examine this basic capability, we propose five subtasks with the difficulty gradually increasing: (1) *Standard*: In this setting, we simplify all settings to form the most basic and standard version of the task, which serves as a foundation for extending to more complex variants. Following this principle, we design the attribute description templates to be as straightforward and uniform as possible across all biographies, and make each sentence in a bio include the person's full name, e.g., " The hobby of Andrew Xavier Jimenez is radio- controlled model collecting.". Each biography consists of six sampled attributes per individual. The task requires the model to retrieve a specific attribute from the context. The *Standard* setting could be regarded as an ameliorated version of NIAH, where the needle is the bio containing the answer, and the haystack is other bios. However, our setting could be more challenging because of improved pertinence between the needle and the haystack. Based upon the *Standard* setting, we gradually add more confounders to increase the difficulty. Specifically, we propose (2) *Multi Standard* to ask the model to simultaneously retrieve $n$ different attributes from $n$ different

Table 2: Task Overview for the LongBioBench. Here {Pn} refers to the name of the n-th person. Acc is the accuracy of the exact match.

| Task | Description | Metric | Example |
|---|---|---|---|
| **Understanding** | | | |
| Standard | Retrieve a specific attribute of one person. | Acc | **Attribute:** The hobby of {P1} is dandyism. 
 **Question:** What's the hobby of {P1}? |
| Multi_standard | Retrieve multiple attributes of different people. | All-or-Nothing Acc | **Attribute:** The hobby of {P1} is dandyism. {P2} is mycology. 
 **Question:** What's the hobby of {P1} and {P2}? |
| Paraphrase | Attribute expressions are paraphrased. | Acc | **Attribute:** {P1} worked in Dhaka. 
 **Question:** Which city did {P1} work in? |
| Pronoun | Bio written from first-person view. | Acc | **Attribute:** I was born on 1993-06-26. 
 **Question:** What is the birthday of {P1}? |
| **Reasoning** | | | |
| Calculation | Compute age difference between two people. | Acc | **Attribute:** {P1} is 61, {P2} is 43. 
 **Question:** What's their age difference? |
| Rank | Rank people by age. | Acc | **Attribute:** {P1} is 61, {P2} is 43. 
 **Question:** Rank from youngest to oldest. |
| Multihop | Retrieve an attribute via cross-person reference. | Acc | **Attribute:** {P1} born in Santa Paula. {P2} born same place as {P1}. 
 **Question:** Birthplace of {P2}? |
| Twodiff | Identify two people with specific age difference. | Acc | **Attribute:** {P1} is 61, {P2} is 43. 
 **Question:** Who has 18 years age difference? |
| **Trustworthy** | | | |
| Citation | Answer plus source citation. | Citation Acc | **Attribute:** Bio [1]: {P1} born in Santa Paula. 
 **Question:** Which university did Isabel graduate from? |
| IDK | No-answer case detection. | Refuse while Answer Acc | **Attribute:** Attribute removed. 
 **Question:** What's the hobby of {P1}? |

persons. The needles are irrelevant to each other and can be located at very different locations in the context. (3) *Paraphrase* provides more diverse templates for attribute description, examining how models capture information with paraphrased templates. (4) *Pronoun* is an updated version of *Paraphrase* by converting the original third-person description to a self-introduction, thus requiring models to understand the pronoun reference better.

**Long Context Reasoning**   Reasoning is another critical skill for models to solve real-world tasks. We design four subtasks to test the model's reasoning ability with a long context. All of them are based on the *Multi Standard* setting. (1) *Rank* asks the model to rank $n$ people according to their age. The task is quite simple if $n$ is small. So we extend it to (2) *Calculation* to ask the model to calculate the age difference of two people, and also (3) *Two-diff*, which requires finding two people with a specific given age difference. Note that the models must plan on what bios to retrieve in the context to solve *Two-diff*, which is also different from previous synthetic benchmarks. As those tasks mainly focus on numerical reasoning, we also present (4) *Multi-hop* tasks where some bios are dependent on each other, and the model needs to figure out the answer by looking through all related bios.

**Trustworthiness**   Beyond understanding and reasoning, we also test the model's trustworthiness in the long-context setting. We propose the following two subtasks: (1) *Citation*: Built upon the *Standard* setting, we index the bios and ask the model to retrieve answers while referring to the bio presenting the target attribute with its index. Therefore, for this task, we not only evaluate the final accuracy but also the precision of the model's citation. (2) *IDK*: Another desirable feature in real-world applications is that the model should refuse to answer the question if the target information is not provided in the context. Therefore, we reuse the same data points from the *Standard* setting and deliberately remove the target information. Considering that weaker models tend to refuse all questions when the task becomes more difficult (e.g, longer context or a harder tasks), we evaluate models with a combination of standard and needle-removed context.

## 3   Main Evaluation Results

### 3.1   Evaluation setup

We evaluate 15 open-source LCLMs supporting more than 128k context lengths, such as Llama [22], Phi [23], Qwen2.5 [24], Mistral [25]. Details are summerized in App. F. We also include three

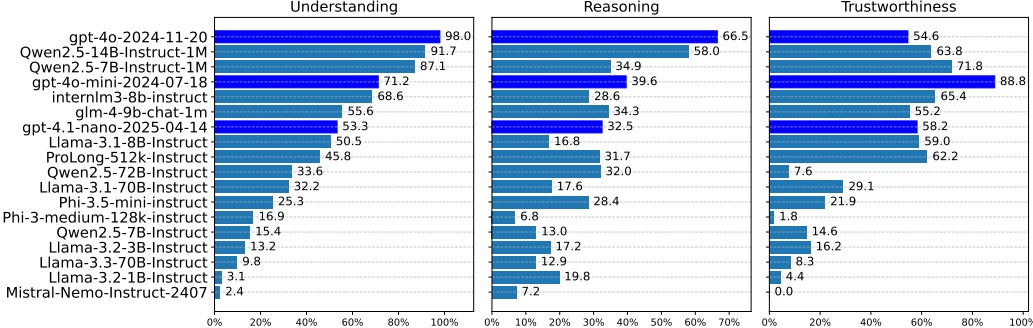

Figure 3: The average performance of all models on Understanding, Reasoning, and Trustworthiness categories.

closed-source models in the GPT family [26]. Each model is evaluated at input lengths: $L \in \{2K, 8K, 16K, 32K, 64K, 128K\}$ where L is the number of tokens counted with the tokenizer of each model[2]. We use zero-shot prompts in all understanding and IDK tasks, and use 2-shot prompting for all reasoning and citation tasks to ensure that the model follows the answer format. We list the prompts for all tasks in App. I. During initial studies, we observed that the model's performance on our proposed tasks nearly converges [3] at around 800 data points, so each test set contains 800 data points. We use the vLLM [27] framework for inference with 8×H800 GPUs. We use greedy decoding for all models. Regarding evaluation metrics, we use accuracy with **exact match** for all understanding and reasoning tasks (all-or-nothing accuracy is applied in the multiple retrieval case). We measure the accuracy of the citations only in the citation task. For the IDK task, the metric is the proportion of questions where the model answers correctly when information is present and refuses to answer when it is absent. We also provide a simple analysis on the hallucination rate in App.G.2.

The overall performance is shown in Fig. 3, and the individual scores for each task are shown in Fig. 4. We draw the following key observations based on the figures.

## 3.2 Validating all proposed tasks

One ideal benchmark for evaluating a model's long-context capability should first fulfill the following two criteria: (1) The tasks should be solvable by the model in a short context. ensuring that model performance decreases mainly because of the change in context length. As indicated in Fig. 5, almost all models achieve near-perfect performance at 2k-token context length on our proposed tasks, except *twodiff*, which was intentionally designed as an example to show how our framework can be extended to more challenging tasks. (2) The tasks ought to be *unsolvable* if the context is unprovided. In certain natural long-context tasks, the model can even achieve reasonable performance without relying on context by just using its parametric knowledge [28]. To validate this point, we test LLaMA-3.1-8B and Qwen2.5-7B with only questions from the standard setting without context. Both models fail the task, ensuring that LongBioBench is not contaminated by memorized knowledge.

## 3.3 Challenges for current long context modeling

This section summarizes six key results observed when evaluating LCLMs on LongBioBench. We first analyze the overall performance across all LCLMs from Fig. 3 (R1 and R2) and look deeper into the performance of different tasks in Fig. 4 (R3-R6). The full results are shown in App. H.

**R1: Open-sourced LCLMs struggle at elementary numerical reasoning and trustworthy tasks, even though they can retrieve.** While some LCLMs demonstrate strong performance on *understanding* tasks, they tend to struggle on *reasoning* and *trustworthiness* ones. A s shown in Fig. 3, GPT-4o, Qwen2.5-14B-1M, and Qwen2.5-7B-1M achieve over 85% accuracy on understanding, but the highest accuracy on reasoning is only 66.5%, and no model exceeds 90% on trustworthy behavior tasks. Notably, most of our reasoning tasks only involve elementary numerical reasoning. So we expect that the model may totally fail the more complex real-world tasks, suggesting substantial room for improvement in these two areas. If we dive deeper into how models fail the reasoning task, we find that the models are capable of retrieving the relevant information. To disentangle retrieval ability from reasoning, we compare performance on reasoning tasks with that on multi-retrieval tasks

---

[2]We did not perform experiments for longer context at scale since most models already perform bad at 128k context length. We provide the results using 256k and 512k context length on three best models in App.G.3

[3]The accuracy fluctuates within 1% for 32 data points

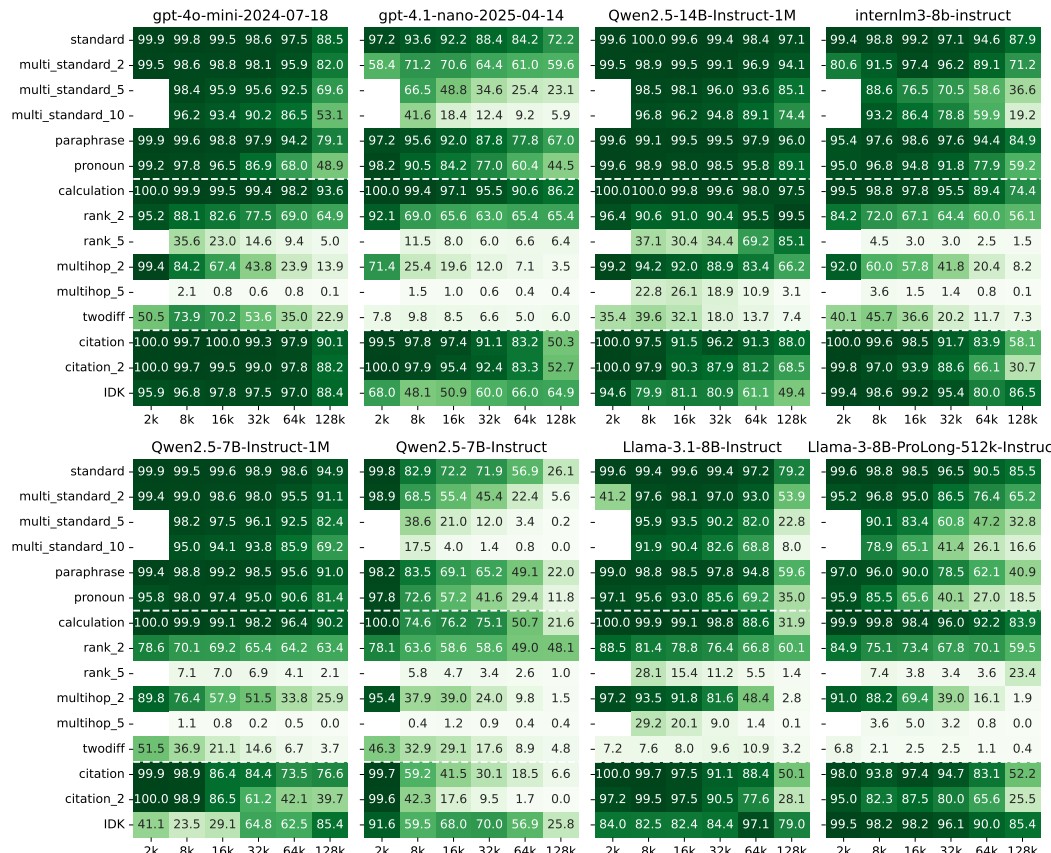

Figure 4: The performance of 8 different models by tasks. Some results are blank since the length of target biographies exceed the specified context length.

in Fig. 4, as the former are deliberately constructed by extending the latter. Across all tasks, we consistently observe a substantial performance gap between multi-retrieval and multi-hop reasoning, indicating that while LCLMs can successfully recall relevant information, they struggle to reason over it effectively. Besides, it is worth noting that for some models, the extended calculation task score is not bounded but exceeds that of the multi-retrieval task. This is because these models are particularly sensitive to retrieving numerical information, allowing them to surpass the expected bounded performance (Discussed in Sec. 4.1).

**R2: Poor Correlation between trustworthiness and task-solving performance.** We cannot find a clear correlation when comparing trustworthiness scores with performance in understanding and reasoning tasks. For example, although GPT-4o achieves the highest scores in both understanding and reasoning, it ranks lower on the citation and IDK. Furthermore, all models exhibit similar trustworthiness performance under the 2k context setting, underscoring the distinct challenge of ensuring safety alignment in long-context scenarios.

**R3: Context length is still the main bottleneck.** As shown in Fig. 4, the performance of all models consistently declines on almost all tasks as the context length grows. Notably, certain models, such as Llama-3.1-8B-Instruct, experience a sharp drop in performance when the context is extended from 64k to 128k, suggesting that the model's effective context length may be shorter than its advertised capacity [6], underscoring that the long-context problem remains an open challenge.

**R4: Poor correlation between calculation and other reasoning tasks** Fig. 4 indicates that most models perform well on simple arithmetic calculations involving the ages of different individuals. However, their performance drops significantly when the task shifts to ranking these ages , even though ranking requires a similar level of numerical reasoning. (Note that the baseline random guess for 2-ranking is 50%) The performance further declines when transitioning from numeric operations to textual comprehension in multi-hop reasoning. These results suggest that while some LCLMs are proficient at numerical calculation, this capability does not generalize to other forms of reasoning.

**R5: LCLMs struggle on constrained planning problem**   We construct twodiff as an example of a hard task for LCLMs. The answer to this task is not unique, and all bios in the context could serve as the needle. As shown in Fig. 4, even models with strong multihop and arithmetic reasoning abilities—such as Qwen-2.5-14B-1M—struggle with constrained retrieval, failing to perform well even at a 2k context length. Besides, none of the models perform over 30% accuracy at 128k context level, which could be easier with a bigger selection pool. This aligns with the findings from [5] that LCLMs only use a small part of context when doing the reasoning task. This highlights a fundamental limitation: current LCLMs remain far from being able to reason effectively over long contexts.

## 4   Analysis

### 4.1   Some LCLMs are more sensitive to retrieving numerical than textual information

Observing Fig. 4, we find that certain LCLMs, such as InternLM3-Instruct and Qwen2.5-7B-Instruct, counterintuitively perform better on the calculation task — an upgraded variant of the 2-retrieval task — than on the 2-retrieval task itself. We hypothesize that this counterintuitive result stems from these models' stronger ability to retrieve and manipulate numerical values than textual information. Since the 2-retrieval task requires extracting a broader range of attribute types, it poses a greater challenge. To investigate this, we cluster each question from the standard setting by attribute types and report the corresponding accuracies in Fig. 5. The figure reveals that InternLM-8B, Prolong-8B, and Qwen2.5-7B achieve their highest scores when retrieving numerical birthdate attributes,

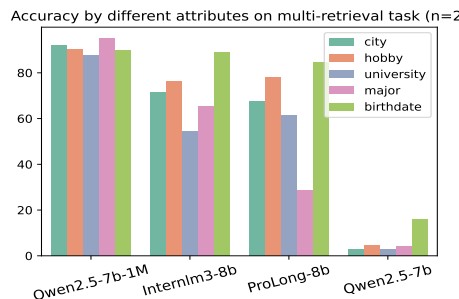

Figure 5: The bar chart of performance by different attributes on the 2-retrieval task. For simplicity, we abbreviate the names of all instruct models.

and all models exhibit stronger performance on the calculation task than the 2-retrieval task. In contrast, Qwen2.5-7B-Instruct-1M demonstrates more balanced performance across all attribute types, and its accuracy on the calculation task is bounded by the 2-retrieval task. These findings support our hypothesis: certain LCLMs appear to be particularly effective at extracting numeric information rather than textual attributes, and this feature appears when their calculation tasks have higher scores than the 2-retrieval tasks.

### 4.2   Coherent context is important

To highlight the importance of contextual coherence and to draw a comparison with NiaH-style tasks, we introduce a bio-in-a-haystack (BiaH) setting. For each task in our benchmark, we construct BiaH by replacing the original context with a haystack in NiaH (we use the Paul Graham essay [4] as the haystack) while preserving the key information relevant to the question. We evaluate the best-performing model, Qwen2.5-Instruct-7B-1M, on both BiaH and our benchmark. The results are presented in Fig. 6. The results reveal a clear performance gap between BiaH and BioBench. While the gap is modest in simpler settings such as standard retrieval (-7.9%), it widens substantially as task difficulty increases, reaching -28.3% and -88.9% in more complex scenarios with larger retrieval scopes. This trend indicates that LLMs exploit incoherent cues as shortcuts when faced with harder tasks. These findings underscore the importance of ensuring context coherence in synthetic contexts.

### 4.3   Performance trend during long-context continual pre-training

**Setting**   To analyze how different capabilities evolve during long-context continual pretraining, we evaluate our benchmark on checkpoints of long-context continual pretraining for Qwen2.5-7B, where the context length is extended from 4,096 to 32,768 tokens [24]. We use the Qwen2.5 checkpoints from 2k to 20k training steps. All tasks are conducted in a 2-shot setting to ensure the model adheres to task instructions, with the reasoning task using chain-of-thought prompts. We only test on 32k

---

[4] `https://github.com/gkamradt/LLMTest_NeedleInAHaystack/tree/main/needlehaystack/PaulGrahamEssays`

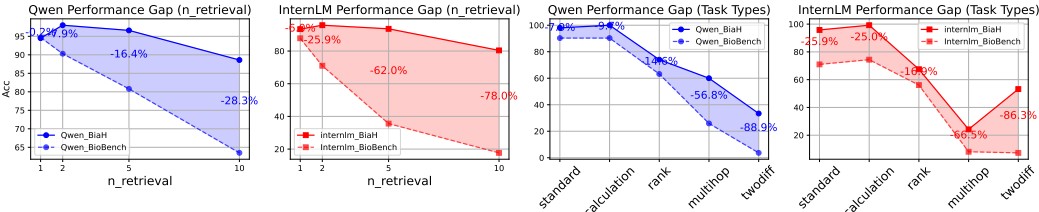

Figure 6: Performance of Qwen-7b-Instruct-1M and InternLM-7B-Instruct on both our LongBioBench and BiaH. The task is controlled as standard retrieval on the left figure,s and the retrieval number is fixed to be 2 on the right figures. A bigger gap is observed on both models as the task difficulty increases.

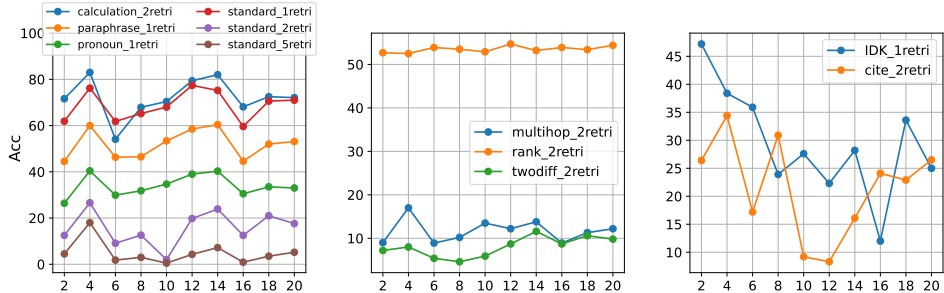

Figure 7: The performance of Qwen2.5 at the long-context pretraining stage on our bench with 32K context length. The x-axis represents the number of training steps. The y-axis shows the accuracy over different tasks.

context lengths on our benchmark to adapt the max context window of the model. The results across all tasks are presented in Fig. 7. Our findings are as follows.

**Performance saturates at the early stage.** Based on the left figure in Fig 7, we observe a significant performance improvement across all tasks in the early training stages. After peaking, the performance slightly declines and then stabilizes with minor fluctuations. This suggests that during the initial 4K training steps, the model rapidly adapts to the previously unseen RoPE embeddings. Notably, accuracy on the retrieval task peaks at the 4K step checkpoint, indicating that a relatively small amount of data may be sufficient to unlock long-context capabilities in LLMs, with additional training yielding marginal gains.

**No noticeable improvements on the reasoning abilities are gained through long-ciontext continual pretraining.** Comparing the middle figure with the left one, we observe that performance improves consistently across all retrieval tasks, whereas the reasoning task shows only a slight improvement, with accuracy remaining extremely low at around 10% (note that the random guess accuracy for the ranking task is 50% since it involves ranking two individuals). Interestingly, the calculation task follows a performance trajectory similar to all retrieval tasks and already achieves high accuracy before long-context pretraining. This suggests that Qwen2.5 already possesses the capability to perform calculations over relatively long contexts, and that the main bottleneck on this task lies in retrieval rather than reasoning. Therefore, we categorize the calculation task alongside the retrieval tasks.

**The model becomes less trustworthy as the training proceeds** The right figure in Fig. 7 shows a consistent decline in performance as pretraining progresses. This suggests that while the model's ability to locate exact information improves with more data, its capability to accurately cite sources and appropriately refuse to answer when information is missing deteriorates. This highlights the necessity of post-training techniques, such as reinforcement learning with human feedback (RLHF), to enhance the model's alignment and reliability in handling uncertain or incomplete information.

### 4.4 Distractor density is another bottleneck for long context tasks

To further investigate the factors influencing the performance of LCLMs, we conduct a stress test on Qwen2.5-Instruct-7B-1M by controlling the **position of the answer information** and the **density of distractors** as the variables while keeping the context length and task fixed. Detailed analysis is shown in App. E and we summarize the results as follows: (1) We observe a strong negative correlation between distractor density and model performance, suggesting that beyond context length,

Table 3: Results across different ICL context lengths.

| ICL | 2k | 8k | 16k | 32k | 64k | 128k |
|---|---|---|---|---|---|---|
| Qwen2.5-14B-Instuct-1M | 51.5 | 25.5 | 30.5 | 25.0 | 16.0 | 17.0 |
| Qwen2.5-7B-Instuct-1M | 20.0 | 19.5 | 11.0 | 11.5 | 10.5 | 12.5 |
| Random Guess | 10.0 | 10.0 | 10.0 | 10.0 | 10.0 | 10.0 |

higher distractor density is a key factor contributing to the difficulty LCLMs face with long-context tasks. (2) We observe the lost-in-the-middle [29] but it is less evident on relatively easier tasks.

### 4.5 In-context Learning (ICL): An Example for Task Extension

As the foundation capabilities of LLMs wiil keep evolving, constructing more creative and challenging reasoning tasks becoming increasingly essential. In this section, We give a simple example on customizing a more challenging task using LongBioBench.

Inspired by [30], which explores long in-context reasoning capabilities, we add a setting similar to the extreme label task: E.g. "Bio1, Bio2. Question: Which category of university did Charlotte Farley Hall graduate from? Answer: Category 3. . . . Bio-n, Bio-n+1 Question: Which category of university did Gabriella Jenson Griffin graduate from?"

In this setting, the context comprises multiple in-context demonstrations, each consisting of several biographies and a corresponding QA pair. For each QA pair, we construct a mapping between university names and categorical labels based on their initial letter (e.g., universities starting with "U" are assigned to Category 1, those starting with "A" to Category 3, etc.). Importantly, this mapping is not explicitly provided to the model. Instead, LCLMs are expected to infer the underlying rule by generalizing from the in-context examples. To prevent LCLMs from exploiting memorization or retrieval of the same university name, we ensure that the queried university is unique. However, to support reasoning, we guarantee that at least one university in the demonstrations shares the same initial character, providing a subtle hint for the mapping.

We perform experiments over 10 uniformly distributed categories on the best model Qwen2.5-7b-Instruct-1M and Qwen2.5-14b-Instruct-1M with a cot prompt. The results are shown in Table 3.

We observe several noteworthy findings regarding the performance of LCLMs under varying context lengths. First, as context length increases, models are provided with more demonstrations to learn from; however, we still find substantial performance drops—for instance, from 8k to 16k tokens for Qwen-7B and from 2k to 8k tokens for Qwen-14B—indicating that in-context learning abilities can be negatively affected when handling longer inputs. Second, even state-of-the-art LCLMs continue to struggle with maintaining robust in-context learning over extended contexts. Thirdly, analysis of failure cases reveals a recurring pattern: models often hallucinate category mappings during their reasoning process despite exhibiting strong retrieval capabilities. This behavior highlights a potentially promising optimization direction for improving current LCLMs, particularly those designed with explicit reasoning mechanisms.

## 5 Conclusion and Limitations

**Conclusion** In this work, we first highlight the limitations of existing long-context evaluation benchmarks: Real-world tasks often lack controllability and are costly to construct, while current synthetic benchmarks frequently overlook the coherence between the inserted information (needles) and the surrounding context (haystacks). We argue that an ideal synthetic benchmark should meet three key criteria: seamlessness, controllability, and soundness. To this end, we introduce LongBioBench, a synthetic benchmark composed of artificial biographies that satisfy all three principles and demonstrate high correlation with existing real-world task benchmarks. Testing 18 LCLMs on these benchmarks in a controllable setting, we found that although current models can retrieve relevant information, they struggle when we extend the task into the reasoning or more complex scenarios. We hope LongBioBench will facilitate more controllable and diagnostic evaluation of LCLMs and serve as a valuable framework for the research community.

**Limitation** We only focus on the most straightforward extension for each task for the controlled study. There will be broad space for extending more challenging tasks, such as in-context learning [30] or passage reranking tasks [11]. We do not focus on more closed-source LCLMs such as Gemini [31], Claude [32] and models using linear attention [33] due to the funding and computation budget. We leave these evaluations as future works.

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

# A    Related Work

With the growing interest in long-context language modeling, various benchmarks have emerged. Some focus on real-world tasks across diverse domains, such as document understanding [3, 34, 28], safety [35], and medical question answering [36]. Others try to collect a collection of real-world tasks aimed at a comprehensive evaluation [2, 37, 4]. However, these benchmarks are often costly to construct and suffer from limited interpretability, as it is challenging to control task complexity systematically. On the other hand, many synthetic tasks are proposed due to their low construction cost and high flexibility [6, 38, 30, 5, 7, 39]. Among them, Needle-in-a-Haystack (NIAH) [19] is the most popular setting, where a "needle" is inserted at a long essay (i.e., the haystack) and the model is tasked with retrieving it. There are also variants of NIAH such as RULER [6]. Nonetheless, several studies have raised concerns about the limitations of this approach [5, 8]. In this work, we argue that the semantic irrelevance between the needle and the surrounding context may allow models to exploit cues or shortcuts, ultimately reducing the challenge and bringing bias into evaluations. We note that OpenAI-MRCR [9] also features the blending between needles and haystacks. However, it is less extensible than LongBioBench as the reasoning pattern required to answer is fixed (e.g find the 2nd poem about tapirs), which can be regarded as a specific instantiation of an extension of LongBioBench.

# B    Dataset Construction Details

We introduce the details of constructing the data in this section: The overall idea is: Firstly, we sample attributes for each person, allowing the benchmark to be generated on the fly and remain independent of any parametric knowledge. Second, we use these attributes to generate coherent biographical text for each individual. Finally, we concatenate multiple bios to construct a full context, where the configuration can be freely adjusted to control the task setup, thus making the framework highly extensible and interpretable.

**Attribute Sampling**    To separate the parametric knowledge within LLMs and enable on-the-fly generation and inspired by the bioS dataset [20], we sample attribute values and corresponding sentence templates uniformly from a collected pool. Specifically, each biography includes seven attributes: *full name, birthdate, birthplace, hobby, graduated university, major, and working city*. We generate 100 unique first, middle, and last names independently using LLaMA-3.1-8B-Instruct and ensure that the resulting full names are unique. Birthdates are sampled uniformly from 1950-01-01 to 2001-12-31. For the other attributes, we extract values from datasets on Kaggle[5], selecting the top 500 most common universities and 300 most common working cities.

**Bio construction**    To generate a coherent bio for each individual, we manually write a clear and straightforward description template for each attribute. These templates are used to construct the biography as a sequence of six sentences (excluding the full name) in the bios construction stage. In the standard setting, all biographies use the same sentence templates, and the attribute order is fixed for consistency.

To support evaluation under more semantically diverse conditions, we also provide various paraphrases for each template generated from LLama-3.1-8b-Instruct. Each paraphrase is manually reviewed to ensure clarity and eliminate ambiguity.

To maintain control over content structure and quality, paraphrasing is applied at the sentence level only, and we retain the original attribute order since variations in order showed a negligible effect on the model performance.

**Context Synthesis**    We use a controllable context construction based on three key configurations: *key information number*, *key information position*, and *distractor density*. Key information number refers to the number of information required to answer the question, and key information position means the position of the question information. Moreover, we introduce an exclusive feature distractor density, which represents the density of the same attribution appears within the context. Our experiments show that the knowledge density can be another strong bottleneck for the long-context

---

[5]https://www.kaggle.com/datasets

tasks. Given those configs, we construct the context in a needle-insertion manner. Specifically, we first construct the haystack by keep concatenating the bios until it reaches a context threshold and then inserting the questioned bios. We use a specific config to control the position where the question bios are inserted. Then we got a sample context and its corresponding question-answer pair.

## C   Data Statistics

We provide the statistics across the average number of biographies in each task in Table 4 and the average token length for all biographies in Table 5.

Table 4: The average number of biographies in a randomly generated Longbiobench dataset.

| Task/Length | 2 | 8 | 16 | 32 | 64 | 128 |
|---|---|---|---|---|---|---|
| standard | 14.93 | 73.77 | 152.19 | 308.83 | 622.18 | 1248.75 |
| paraphrase | 12.88 | 62.36 | 128.01 | 263.86 | 528.61 | 1060.80 |
| pronoun | 15.07 | 75.21 | 156.65 | 316.81 | 636.10 | 1276.81 |
| multi_standard | 12.05 | 70.87 | 149.28 | 305.35 | 617.94 | 1246.20 |
| calculation | 12.84 | 76.34 | 161.12 | 330.70 | 668.71 | 1348.31 |
| rank | 12.86 | 75.77 | 160.62 | 329.90 | 667.42 | 1345.70 |
| multihop | 12.06 | 70.88 | 149.27 | 305.82 | 619.39 | 1246.22 |
| twodiff | 12.85 | 76.31 | 161.42 | 330.84 | 670.33 | 1347.75 |

Table 5: The average number and standard deviation of tokens within each biography tokenized by Qwen2.5-7b-Instruct.

| Task/Length | 2 | 8 | 16 | 32 | 64 | 128 |
|---|---|---|---|---|---|---|
| standard | $105.65_{8.35}$ | $105.42_{8.40}$ | $105.45_{8.41}$ | $105.52_{8.39}$ | $105.54_{8.26}$ | $105.57_{8.31}$ |
| paraphrase | $126.85_{12.68}$ | $125.16_{12.15}$ | $125.59_{11.60}$ | $123.48_{11.45}$ | $124.14_{11.23}$ | $124.14_{11.21}$ |
| pronoun | $103.13_{5.89}$ | $103.25_{7.33}$ | $102.37_{7.84}$ | $102.90_{7.73}$ | $103.25_{7.60}$ | $103.28_{7.46}$ |
| calculation | $97.69_{8.41}$ | $97.58_{8.32}$ | $97.61_{8.43}$ | $97.62_{8.42}$ | $97.78_{8.46}$ | $97.61_{8.41}$ |
| multihop | $106.32_{8.56}$ | $105.74_{8.47}$ | $105.65_{8.44}$ | $105.65_{8.36}$ | $105.57_{8.45}$ | $105.56_{8.32}$ |
| multi_standard | $105.72_{8.31}$ | $105.66_{8.52}$ | $105.56_{8.52}$ | $105.77_{8.38}$ | $105.80_{8.43}$ | $105.56_{8.31}$ |
| twodiff | $97.63_{8.29}$ | $97.63_{8.32}$ | $97.44_{8.33}$ | $97.57_{8.39}$ | $97.55_{8.36}$ | $97.64_{8.41}$ |
| rank | $97.45_{8.46}$ | $97.65_{8.41}$ | $97.60_{8.42}$ | $97.70_{8.46}$ | $97.90_{8.60}$ | $97.76_{8.43}$ |

## D   Task Description

This subsection will outline our motivation and explain how we developed all the current tasks in the proposed bench.

The tasks are split into three categories: understanding, reasoning, and trustworthiness, representing the core capabilities required to solve the tasks. Detailed examples of the benchmark are presented in Table 2.

**Standard Information Retrieval (Standard).**   We start with the simplest retrieval settings as the *Standard* version. To allow for increments in task difficulty, we ensure that all statements are expressed using the simplest and most direct sentences, such as " The hobby of *{person}* is ". This also avoids ambiguity for models, which establishes a robust baseline for subsequent, more challenging tasks. The model will be asked to retrieve a specific attribute for a person.

**Multi Information Retrieval (Multi_standard).**   To further challenge the model to simultaneously retrieve information across different context locations, we upgrade the single retrieval task to a multi-retrieval task by asking models to retrieve n attributes from n people instead of one, where n can be modified when constructing the dataset. Here we let n equal 2, 5, 10 by default.

**Retrieval on Paraphrased Bios (Paraphrase).**   To demand stronger contextual understanding, we paraphrase the expression of attributes within the bios. This prevents models from relying on exact matches between questions and sentences to locate answers. As a result, we can control for other confounding factors and more accurately assess the models' true comprehension capabilities by examining the performance gap relative to the *Standard* version.

**Retrieval on Bios stated with Pronoun (Pronoun).**   This task is an extension of the paraphrasing task. Based on the paraphrase setting, each bio is rewritten as a self-introduction. All sentences that describe a person's attributes are expressed in the first person, with the individual's name appearing only at the beginning of the bio. This design builds upon sentence-level understanding in paraphrasing and further challenges the LLM's ability to understand the paragraph-level semantics, which is the hardest task in the understanding category.

**Calcuating the Ages (Calculation).**   For the reasoning level, we require the LLM to reason on the retrieved information. The calculation task asked LLM to calculate the subtraction of the ages of two people. We use subtraction here instead of the summation to make this task expandable to the TwoDiff task later. Besides, to prevent the ages of people from changing over time, we note that all birthdate attributes are replaced by the specific ages under this setting.

**Ranking the Ages (Rank).**   We extend the Calculation setting to ranking the ages of different people so that we can freely define the number of retrieved information by specifying the number of people to be ranked. Here we let n equal 2 and 5 by default since we observe that 5 retrievel ranking task is challenging enough for most models.

**Retrieve Two People Satisfying the Age Difference (Twodiff).**   In this task, we give LLM an age difference and ask LLM to retrieve two people whose age difference satisfies the age difference. This demands that LLMs plan on retrieving the target instead of directly retrieving it based on the given information. We design this task as a naive simulation of the scenario where LLMs are asked to do some constrained retrieval (e.g in pairs trading, traders look for two stocks whose price difference equals a predetermined target).

**Multi-hop Retrieval (Multihop).**   Multi-hop question answering is a popular setting in document question answering. In our benchmark, we replicate this setting by randomly changing the expression of an attribute into "The {attribute} of {person 1 name} is the same as {person 2 name}" where we ensure that person 2 appeared after person 1 in the context. This forces LLM to understand the expression and retrieve sequentially across different positions in the context, which is an extended, harder version of multi-retrieval.

**Citation (Cite).**   Built upon the *Standard* setting, we index the bios and ask the model to retrieve answers while referring to the bio presenting the target attribute with its index. Therefore, for this task, we not only evaluate the final accuracy but also the precision of the model's citation. Generating with citations has long been an essential ability of trustworthy LLMs. We test their capabilities on citing the correct bios after their answer in this task. To make the citation trackable, we add a number before each bio and ask LCLM to generate both the answer and its corresponding number and measure the accuracy of the citation in the end. As an extended version of *Standard/Multi_standard*, we set the number of information pieces to 1 and 2 by default.

**I don't know (IDK).**   Expressing uncertainty is a critical aspect of trustworthy behavior in LLMs [40]. To evaluate this, we simulate a controlled setting in which the target information is deliberately removed, and the LLM is prompted to respond with "The answer is not explicitly stated." Observing that weaker LLMs tend to refuse all questions when the task becomes more difficult (e.g, with longer context or a harder version), we evaluate models based on a combination of standard retrieval and uncertainty expression. Specifically, a model is considered to have successfully passed a question only if it (1) correctly retrieves the attribute when the relevant information is present, and (2) appropriately refuses to answer when the attribute sentence is removed.

## E   Analysis: Density and Needle position v.s Performance

To further investigate the factors influencing the performance of LCLMs, we conduct a stress test on Qwen2.5-Instruct-7B-1M by controlling the **position of the answer information** and the **density of distractors** as the variables while keeping the context length and task fixed. Specifically, we adjust the percentage of the haystack depth to insert the needle for controlling the position and set the probability of generating the same attribute as the needle attribute to control the distractor density. We evaluate the model on two representative tasks: the simplest reasoning task, *calculation*, and the most challenging understanding task, *pronoun*, as their baseline performances lie in a moderate range—neither too high nor too low. The results are visualized in the heatmap shown in Fig. 8.

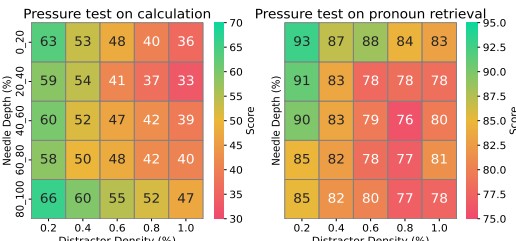

Figure 8: The performance on calculation (left) task and pronoun retrieval (right) task corresponding with the answer depth and distractor density. Y-axis shows the percentage of insertion depth in the context and x-axis shows the percentage of the distracted information appeared in the context.

Our key observations from the figure are as follows. We first observe a strong negative correlation between distractor density and model performance, suggesting that beyond context length, higher distractor density is a key factor contributing to the difficulty LCLMs face with long-context tasks. Second, we observe the lost-in-the-middle [29] phenomenon with our proposed synthetic task *calculation*, where performance declines when the needle appears in the middle of the context. Interestingly, this trend is less evident in the *pronoun* task. We conjecture that this is because the model already performs relatively well on the *pronoun*. Finally, both of these effects—performance decay with density and positional sensitivity—are more pronounced in the reasoning task than in the understanding task. This suggests that certain failure patterns emerge only under sufficiently challenging conditions, reinforcing the need to continue developing more difficult long-context benchmarks.

## F   Model Details

We provide the details of all models evaluated in Table 6.

Table 6: Details of all evaluated Long-Context Language Models

| Models | Release Date | Size | Support Context Length |
|---|---|---|---|
| gpt-4.1-nano-2025-04-14 | 2025-04 | - | 128,000 |
| gpt-4o-2024-11-20 | 2024-11 | - | 128,000 |
| gpt-4o-mini-2024-07-18 | 2024-07 | - | 128,000 |
| internlm3-8b-instruct | 2025-01 | 8B | 131,072 |
| Qwen2.5-7B-Instruct-1M | 2025-01 | 7B | 1,010,000 |
| Qwen2.5-14B-Instruct-1M | 2025-01 | 14B | 1,010,000 |
| Llama-3.3-70B-Instruct | 2024-12 | 70B | 131,072 |
| Llama-3-8B-ProLong-512k-Instruct | 2024-10 | 8B | 524,288 |
| Qwen2.5-7B-Instruct | 2024-09 | 7B | 131,072 |
| Qwen2.5-72B-Instruct | 2024-09 | 72B | 131,072 |
| Llama-3.2-1B-Instruct | 2024-09 | 1B | 131,072 |
| Llama-3.2-3B-Instruct | 2024-09 | 3B | 131,072 |
| Phi-3.5-mini-instruct | 2024-08 | 4B | 131,072 |
| Llama-3.1-8B-Instruct | 2024-07 | 8B | 131,072 |
| Llama-3.1-70B-Instruct | 2024-07 | 70B | 131,072 |
| Mistral-Nemo-Instruct-2407 | 2024-07 | 12B | 131,072 |
| glm-4-9b-chat-1m | 2024-06 | 9B | 1,048,576 |
| Phi-3-medium-128k-instruct | 2024-05 | 14B | 131,072 |

## G   Further Analysis

We provide some further analysis in this section.

## G.1 Further analysis on the task correlation with HELMET

To further understand which setting in long-biobench is closer to Helment, we perform an analysis on the correlation of each subtask between our benchmarks and the helmet:

The two-diff task and rank task are the task that have least correlation with helmet. This suggests that the two-diff tasks—which provide an open-ended setting with multiple correct answers and require constrained planning—may represent a new direction that Helmet struggles to evaluate. As for the rank tasks, the difficulty may lie in their simplicity; with random guessing yielding around 50% accuracy. That makes it challenging to differentiate model performance effectively.

| Task | Spearman correlation | p-value |
|---|---|---|
| standard | 0.92 | 0.00 |
| multi_standard | 0.76 | 0.00 |
| paraphrase | 0.87 | 0.00 |
| pronoun | 0.86 | 0.00 |
| rank | 0.27 | 0.39 |
| calculation | 0.83 | 0.00 |
| twodiff | 0.21 | 0.50 |
| multihop | 0.78 | 0.00 |
| citation | 0.70 | 0.01 |
| IDK | 0.57 | 0.05 |

Table 7: Spearman correlations and significance levels for different tasks.

## G.2 Hallucination Rate

Hallucination rate has long been an important metric for measuring the safety of LLMs. Here we report the hallucinated rate on three best models on the single retrieval test. If the output is failed against the golden label and is not found in the context, we treat this output to be hallucinated. The hallucination rate is determined by dividing the number of hallucinated cases by the total number of failed cases. From the results in Table 8, we found that LCLMs, especially for some strong models, are still suffer from hallucinations.

Table 8: Hallucination statistics across models.

| Model | Hallucinated rate (%) | Hallucinated / Failed / Total |
|---|---|---|
| Qwen2.5-14B-Instruct-1M | 58.8% | 14 / 24 / 800 |
| Qwen2.5-7B-Instruct-1M | 9.5% | 4 / 42 / 800 |
| InternLM3-8B-Instruct | 23.5% | 32 / 98 / 800 |

## G.3 Experiment Results extending to 512k Context

As we already observe a significant performance drop in the range from 2k to 128 context length. We did not perform experiments at scale with further extended context length for all models. We tested the SOTA models qwen2.5-7b-1M and qwen2.5-14b-1M on 256k and 512k context lengths. The results are shown in Table 9.

Table 9: Accuracy (%) across different models and context lengths.

| Context length | Qwen2.5-7B-1M | | Qwen2.5-14B-1M | |
|---|---|---|---|---|
| | 256k | 512k | 256k | 512k |
| understanding | 48.0 | 0.3 | 47.0 | 2.7 |
| reasoning | 37.2 | 31.5 | 40.3 | 28.8 |
| reasoning (w/o rank) | 6.75 | 0.0 | 12.25 | 2.25 |
| trustworthiness | 33.0 | 0.3 | 38.7 | 2.3 |

The performance drops significantly from 256k to 512k and approaches zero at the 512k level. Demonstrating that a context exceeding 512k is still a challenging setting. Despite the poor performance on other tasks, those two models still perform really well at the ranking tasks (almost 100% for 2-rank and having 64% accuracy on 5-rank for qwen14b). This suggests that the model remains effective in handling numerical features within a long context, as is detailed in Sec 4.1. We performed experiments on 1M context and found that all metrics reduce to zero. Interestingly, for the single retrieval task, we found that most of the time, Qwen2.5-7b-1M tends to output 'birthday' regardless

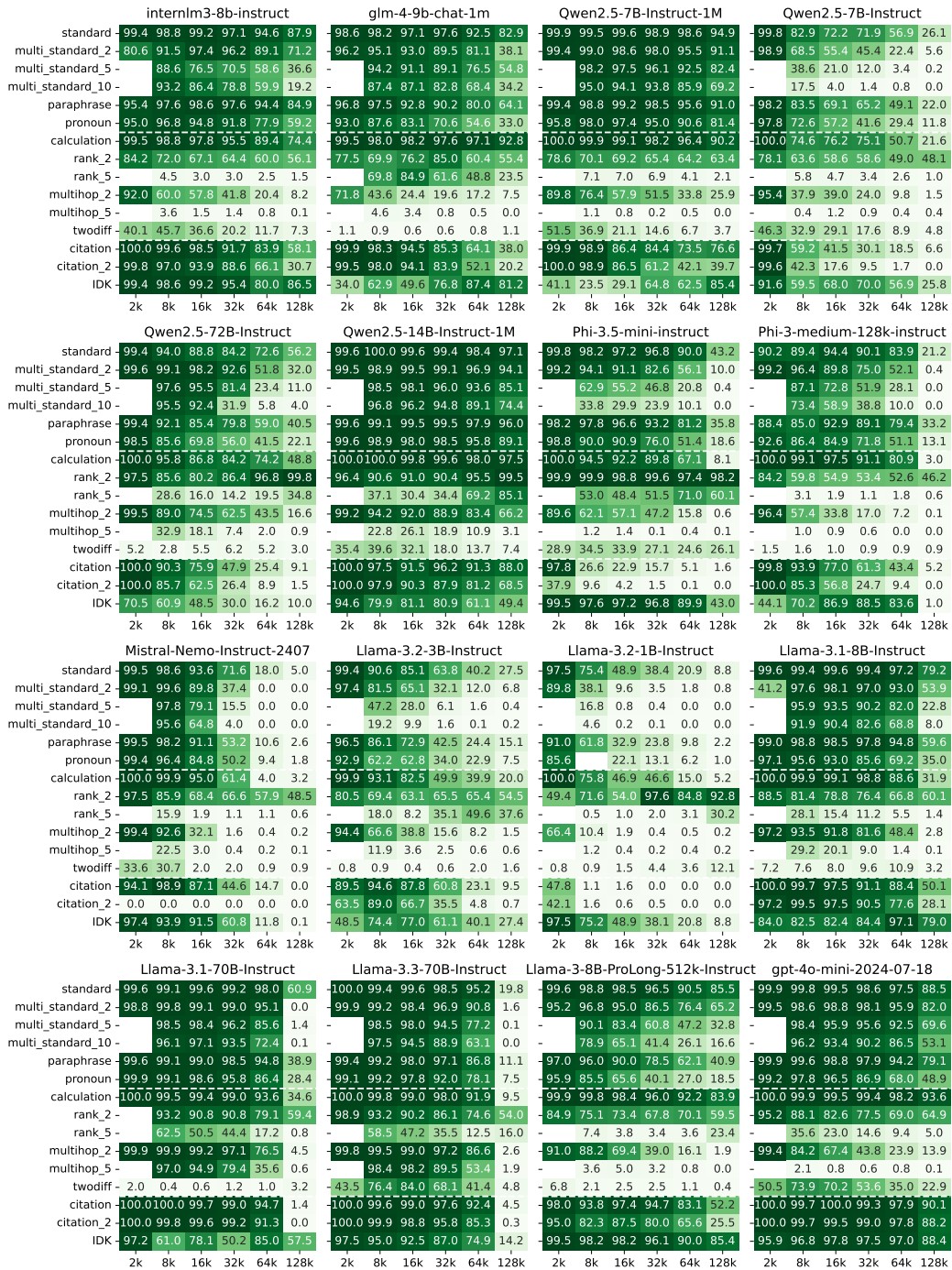

Figure 9: The performance of 16 models by tasks.

of what we ask it. This potentially suggests that the main failure patterns may be too sensitive to numbers within the extremely long context.

# H  Full Results

The full results are shown in Fig. 9.

# I  Prompts

The prompt we used for each tasks are shown as follows:

> **Standard/Paraphrase/Pronoun:**
> **(System)**: Your task is to answer the user's question based on a long context, which consists of many bios. Output the answer only. Don't explain or output other things.
> **(User)**:"Context: {given_context}
> Question: {question}
> **(Assistant)**: Based on the provided context, {question_prefix}

> **Multi_Standard:**
> **(System)**: Your task is to answer all the user's questions based on a long context, which consists of many bios. Output only the answers for each question sequentially. Don't explain or output other things.
> **(User)**: Context: {given_context}
> The Questions are as follows:
> question
> Answer each question in sequence.
> **(Assistant)**: Based on the provided context, the answer is

> **Rank:**
> **(System)**: Following the format of the examples, your task is to rank the users based on their bios in a long context.
> **(User)**: Context: {given_context}
> examples_with_cot
> Question: {question}
> **(Assistant)**: Based on the provided context,

> **Calculation:**
> **(System)**: Your task is to calculate the age difference of the given people based on the given instruction from a long context containing multiple bios.
> **(User)**: Context: {given_context}
> examples_with_cot
> Question: {question}
> **(Assistant)**: Answer: Based on the provided context,

> **Multihop:**
> **(System)**: Following the format of the examples, your task is to answer the user's question based on a long context, which consists of many bios.
> **(User)**: Context: {given_context}
> examples
> Question: {question}
> **(Assistant)**: Answer: Based on the provided context,

**Twodiff:**
**(System)**: Your task is to find the names of people based on the given instruction from a long context containing multiple bios. Follow the format provided in the examples closely and give the final answer.
**(User)**: Context: {given_context}
examples
Question: {question}
**(Assistant)**: Answer:

**Cite (Standard):**
**(System)**: Your task is to answer the user's question with citation based on a long context, which consists of many bios. You must output the answer following with the citation number of the relevant bios strictly surrounded by square brackets such as [1]. Don't explain or output other things.
**(User)**: Context: {given_context}
examples
Question: {question}
Answer:
**(Assistant)**: Based on the provided context, {question_prefix}
**Cite (Multi-Standard):**
**(System)**: Your task is to answer all the user's questions with citation based on a long context, which consists of many bios. Following the format of the examples, You must output the answer ending with the citation number of the relevant bios strictly surrounded by square brackets such as [1]. You should give the answer and citation for each question sequentially. Don't explain or output other things.
**(User)**: Context: {given_context}
examples
question
Answers:
**(Assistant)**: Based on the provided context,

**IDK:**
**(System)**: Your task is to answer the user's question based on a long context, which consists of many bios. Output the answer only. If you don't know the answer or the answer is not explicitly stated, you should strictly output 'The answer is not explicitly stated'. Don't explain or output other things.
**(User)**: Context: {given_context}
Question: {question}
**(Assistant)**: Based on the provided context, {question_prefix}

