# OpenReview forum: "A Controllable Examination for Long-Context Language Models"
_NeurIPS.cc/2025/Datasets_and_Benchmarks_Track — NeurIPS 2025 Datasets and Benchmarks Track spotlight_

### Official Review · Reviewer_jv1z · 2025-06-14

**Rating:** 5
**Confidence:** 4

**Summary:**

The paper introduces LongBioBench, a synthetic benchmark for evaluating long-context language models (LCLMs). It emphasizes seamless context, controllability, and soundness, addressing limitations of existing real-world and synthetic benchmarks. Evaluations of 18 LCLMs reveal struggles in semantic reasoning and trustworthiness, with long-context pretraining showing marginal capability improvements. LongBioBench better balances authenticity and controllability, correlating strongly with real-world tasks, and highlights context coherence and distractor density as key evaluation factors.

**Dataset Code Accessibility:**

Yes

**Ethical Considerations:**

No, there are no or only very minor ethics concerns

**Final Justification:**

All my concerns have been resolved.

**Limitations Weaknesses:**

1. The paper highlights the strong correlation between LongBioBench and helmet benchmark, I am curious about the correlation between previous synthetic tasks (e.g., needle-in-a-haystack, Nolima). Additionally, how does LongBioBench correlate with other real-world long-context benchmarks such as LongBench-v2 and L-Eval? The authors could further explain the underlying reasons for the high correlation between LongBioBench and helmet benchmark?

2. With ultra-long context windows becoming an emerging trend in large language models (many current models already support 1M-token contexts), I'm interested in understanding how different models perform on LongBioBench when evaluated at the 1M context window scale.

**Strengths Contributions:**

1. Three core task categories were synthesized: understanding, reasoning, and trustworthiness, enabling richer and more accurate evaluation of large models' long-context capabilities.


2. Observing performance trends during long-context continual pretraining yielded intriguing findings, providing the open-source community with critical insights to optimize pretraining strategies and address bottlenecks in model reasoning and trustworthiness.

---

> ### Author Rebuttal · Authors · 2025-07-31
>
> Thank you for your review and the support for our comprehensive task categories and analysis for the pretraining. Please let us know if we have addressed your concerns.
>
> > Correlation with other synthetic and real-world benchmarks.
>
> We agree that correlation with other benchmarks helps highlight the difference among benchmarks.
> *Correlation between previous synthetic task:*
>
> - NIAH: Since most models can achieve 100% on NIAH, it’s not straightforward to calculate the correlations with longbiobench.
> - Nolima: Because Nolima is primarily tested on closed-source models, and we mainly test the open-sourced models, there are three models that overlap and thus it’s meaningless to calculate the correlation. We will try our best to evaluate more closed-source models once we have more credits.
>
> *Correlation with other real-world long-context benchmarks*
> - LongBenchV2: We have 7 models that overlap with LongBench - v2. The Spearman correlation is 0.357 with a p-value of 0.43. The results are not significant due to the fact that the performance of longbench-v2 is not distinguished (all performances lie in 29% to 46%)
>
> | Model                     | long-bench-v2 | longbiobench |
> |--------------------------|------------|---------------|
> | gpt-4o-2024-11-20        | 46.00      | 73.03         |
> | Llama 3.3 70B            | 29.80      | 10.33         |
> | Llama-3.1-8B-Instruct    | 30.00      | 42.10         |
> | Llama-3.1-70B-Instruct   | 31.60      | 26.30         |
> | glm-4-9b                 | 30.20      | 48.37         |
> | Qwen2.5-7B-Instruct      | 35.60      | 17.67         |
> | Qwen2.5-72B-Instruct     | 42.10      | 24.40         |
>
> - L-Eval: This benchmark evaluated some early models (e.g LLama2, glm3)  thus we did not have any overlapped models
> Underlying reasoning for high correlation with helmet:
>
> To further understand which setting in longbiobench is closer to Helment, we perform an analysis on the correlation of each subtask between our benchmarks and the helmet:
>
>
> | Task           | Spearman correlation | p-value |
> |----------------|----------|------------|
> | standard       | 0.92     | 0.00       |
> | multi_standard | 0.76     | 0.00       |
> | paraphrase     | 0.87     | 0.00       |
> | pronoun        | 0.86     | 0.00       |
> | rank           | 0.27     | 0.39       |
> | calculation    | 0.83     | 0.00       |
> | twodiff        | 0.21     | 0.50       |
> | multihop       | 0.78     | 0.00       |
> | citation       | 0.70     | 0.01       |
> | IDK            | 0.57     | 0.05       |
>
> The results show that all understanding tasks and some fundamental reasoning task (e.g., calculation, multi-hop) contribute most to the high correlation. And some subtasks, like twodiff and IDK, do not correlate well with the helmet, indicating that the helmet may not have direct indicators to evaluate the model’s trustworthiness behavior.
>
> We agree with the reviewer and fully understand that the differences and overlaps with different benchmarks are valuable. However, we are unable to perform many experiments on closed-source models within the limited rebuttal time. We will add those results in the camera-ready version.
>
> >1M experiments (qwen2.5-7b-1M)
>
> We agree that experiments on models supporting larger context windows may be interesting. We tested the SOTA models qwen2.5-7b-1M and qwen2.5-14b-1M on 256k and 512k context lengths. We note that we found the performance is already near 0 for almost all tasks, so we did not test 1000k context windows at scale. The results and findings are shown as follows: (We will incorporate the detailed results in the camera-ready version.)
>
> | Acc (%)      | Qwen2.5-7B-1M (256k) | Qwen2.5-7B-1M (512k) | Qwen2.5-14B-1M (256k) | Qwen2.5-14B-1M (512k) |
> |---------------|----------------------|-----------------------|------------------------|------------------------|
> | understanding | 48.0                   | 0.3                   | 47.0                     | 2.7                    |
> | reasoning     | 37.2                 | 31.5                  | 40.3                   | 28.8                   |
> | reasoning (w/o rank)    | 6.75                 | 0                  | 12.25                   | 2.25                   |
> | trustworthiness   | 33                   | 0.3                   | 38.7                   | 2.3                    |
>
>
> The performance drops significantly from 256k to 512k and approaches zero at the 512k level. Demonstrating that a context exceeding 512k is still a challenging setting.
> Despite the poor performance on other tasks, those two models still perform really well at the ranking tasks (almost 100% for 2-rank and having 64% accuracy on 5-rank for qwen14b). This suggests that the model remains effective in handling numerical features within a long context, as is detailed in Sec 4.1.
> We performed experiments on 1M context and found that all metrics reduce to zero. Interestingly, for the single retrieval task, we found that most of the time, Qwen2.5-7b-1M tends to output 'birthday' regardless of what we ask it. This potentially suggests that the main failure patterns may be too sensitive to numbers within the extremely long context.

---

> > ### Comment · Reviewer_jv1z · 2025-08-05
> >
> > Thank you for your reply! My concerns have been addressed, and I have accordingly increased the rating to 5.

---

> ### Comment · Area_Chair_MMa9 · 2025-08-04
> **Please engage with authors during the discussion period**
>
> Dear reviewer jv1z,
>
> The authors have responded to your review with a rebuttal that addresses the two limitations that you noted. Please review the rebuttal that they have posted and respond to indicate whether your concerns have been addressed, or, if not, what information the authors should provide or what changes they should make to address them, to the degree that that is possible. There are only a couple of days left in the discussion period, and your engagement during this time is critical.

---

### Official Review · Reviewer_wJaf · 2025-07-02

**Rating:** 5
**Confidence:** 4

**Summary:**

Inspired by Zeyuan Zhu, this paper proposed LongBigBench for evaluating long-context language models using artificially generated biographies. The authors critique existing benchmarks: real-world tasks (e.g., HELMET, L-Eval) are costly and suffer from data contamination, while synthetic tasks (e.g., NIAH, RULER) lack coherence between "needles" (target information) and "haystacks" (context). LongBioBench addresses these gaps by prioritizing seamless context, controllability, and sound evaluation.

**Dataset Code Accessibility:**

Partly

**Dataset Code Comments:**

The dataset is generated by LLMs

**Ethical Considerations:**

No, there are no or only very minor ethics concerns

**Final Justification:**

Thanks for your response. Most of my concerns are well addressed. So I decide to change rating to Accept.

**Limitations Weaknesses:**

- LongBioBench's tasks, particularly numerical reasoning, are overly simplified (e.g., age calculations) and rely on templated biographies, which lack the ambiguity and complexity of natural text.
- The emphasis on "seamless context" may inadvertently introduce biases, such as models leveraging narrative predictability rather than true comprehension. While the paper shows that incoherent needles (e.g., NIAH) are easier to retrieve, it does not fully disentangle whether the observed performance gaps stem from coherence or other latent factors (e.g., syntactic cues). Further analysis of failure modes could strengthen the claim
- The trustworthiness evaluation is limited to citation accuracy and IDK (refusal-to-answer) scenarios, omitting other critical aspects like factual consistency, hallucination rates, or adversarial robustness in long-context settings.
- The reproducibility of LongBioBench's results i constrained by its synthetic data generation methodology. While the paper provides detailed templates and attribute pools for biography generation, the actual dataset construction relies on LLM-based generation without disclosing: (1) the specific LLMs used for generation, (2) the random seeds controlling sampling variability, or (3) the complete prompt templates governing output formatting. How to ensure the result can be reproduce or how to make it stable?

**Strengths Contributions:**

- Code/data are released; detailed experimental setup (prompts, hyperparameters, hardware).
- This paper is easy to read and follow. There are several interesting observations.

---

> ### Author Rebuttal · Authors · 2025-07-31
>
> Thank you for your review and the summary of our features. Please let us know if we have addressed your concerns.
>
> > Numerical reasoning task is too simple.
>
> We agree with the reviewer that the age calculations may be overly simplified for simulating a real-world long-context reasoning task. However, one of our primary motivations is to provide a controllable and extensible framework for probing long-context models. By progressively increasing the complexity of the task step by step, we can understand their boundaries.
>
> **We therefore begin with the most straightforward task and progressively increase the complexity**. Specifically, for each task category (Understanding, Reasoning, or Trustworthiness), we start with the most standard setting, allowing it to be further extended for a controllable comparison. For example, the two-diff task is directly extended from the calculation settings, allowing us to measure the constraint planning capabilities of LLMs (Fig. 2).
>
> **We stopped proposing more complex problems when we found that most long-context models struggle with the proposed tasks**. We found that most current open-source models and some closed-source models, such as GPT-4o-mini, achieve only 39.6% average accuracy on reasoning tasks despite their strong retrieval performance (73% average accuracy). This highlights the difficulty LLMs face in jointly performing reasoning and retrieval (Results 1, L221). Among open-source models, the best-performing one—Qwen2.5-14B-1M—achieves an average accuracy of just 58% on reasoning tasks, with all other open-source models falling below 50%.
>
> **To demonstrate the extensibility of our proposed framework, we add a new in-context learning setting as an example.** The extensibility here means that the difficulty of the tasks can be **flexibly adjusted**. We also enhance the complexity of the templates by introducing paraphrase and pronoun settings from the standard tasks.
> And to further increase the complexity of the benchmark, we add a long-context In-Context Learning setting. A data point from this task is organized as follows:
>
> E.g. “{Bio1, Bio2}.
> Question: Which category of university did Charlotte Farley Hall graduate from?
> Answer: Category 3.
> …
> {Bio-n, Bio-n+1}
> Question: Which category of university did Gabriella Jenson Griffin graduate from?”
>
> In this setting, the context comprises multiple in-context demonstrations, each consisting of several biographies and a corresponding QA pair. For each QA pair, we construct a mapping between university names and categorical labels based on their initial letter (e.g., universities starting with “U” are assigned to Category 1, those starting with “A” to Category 3, etc.). Importantly, this mapping is not explicitly provided to the model. Instead, LCLMs are expected to infer the underlying rule by generalizing from the in-context examples. To prevent LCLMs from exploiting memorization or retrieval of the same university name, we ensure that the queried university is unique. However, to support reasoning, we guarantee that at least one university in the demonstrations shares the same initial character, providing a subtle hint for the mapping.
>
> We perform experiments over 10 uniformly distributed categories on the best model Qwen2.5-7b-Instruct-1M and Qwen2.5-14b-Instruct-1M with a cot prompt. The results are as follows:
>
> | Accuracy (%)           | 2    | 8    | 16   | 32   | 64   | 128  |
> |---------------|------|------|------|------|------|------|
> | Qwen2.5-14B-Ins-1M | 51.5 | 25.5 | 30.5 | 25   | 16   | 17   |
> | Qwen2.5-7B-Ins-1M | 20   | 19.5 | 11   | 11.5 | 10.5 | 12.5 |
> |Random Guess|10|10|10|10|10|10|
>
> We observe the following interesting findings:
> As the context length increases, we provide more demonstrations for LCLMs to learn from. However, we still observe a significant performance drop from 8k -> 16k for Qwen7b and 2k -> 8k for Qwen14b. This suggests that the in-context learning abilities can still be negatively impacted when LCLMs handle longer contexts.
> Even current SOTA LCLMs still struggle with in-context learning over long inputs
> Observing the failure cases, we found an interesting pattern where the model hallucinates the category mapping it found through their thinking process, even though they have strong retrieval abilities, which could be a promising optimization direction for current LCLMs (especially for those thinking models)
>
> > Introducing bias with seamless context when comparing with NIAH
>
> We are unsure if we fully understand the reviewer’s concern.
> Our understanding for this review is that we show that incoherently inserted needles are easier to retrieve compared with needles in a coherent context. However, this could be due to other latent factors, such as syntactic cues (e.g., for NIAH, there are cues like "The magic number is …").
>
> Our response is: We believe this is a fair comparison, as LongBioBench utilizes similar synthetic cues as in NiAH (“The person is graduated from…”). In both settings, LLMs can leverage those cues. In Fig. 6, we only vary the needle and question. We thus believe the bias introduced by the syntactic cues may not have a significant impact on the results.
>
>
> > Other trustworthy settings are limited to citation and IDK
>
> We agreed that other critical aspects, like factual consistency and hallucination rates, are good measurements. We did not add them to the current framework since they are more like measurement metrics instead of a specific task. It is possible to add the hallucination rates as a new metric.  Here we report the hallucinated rate on three best models. If the output is failed and is not found in the context, we treat this output to be hallucinated.
>
> | Model       | Hallucinated rate (%) | Hallucinated / Failed / Total |
> |-------------|--------------|------------------|
> | Qwen2.5-14B-Instruct-1M    | 58.8%        | 14 / 24 / 800       |
> | Qwen2.5-7B-Instruct-1M     | 9.5%         | 4 / 42 / 800          |
> | InternLM3-8b-Instruct    | 23.5%        | 32 / 98 / 800         |
>
> From the results, we found that LCLMs, especially for some strong models, are still suffer from hallucinations.
>
> Due to the response time limit we are not able to include all the experiments here. We will provide those detailed metrics in the camera-ready version.
>
>
> > Reproducibility Concerns
>
> We believe that there are misunderstandings. Our benchmark can be **deterministically** generated with the prepared templates and fixed seed without any LLM.
>
> For (1), we use LLMs (Llama3.1-8b-Instruct) only for generating the paraphrased templates before constructing the benchmark. These templates are then checked by humans and fixed to generate the benchmark. All the templates used for generating the dataset are submitted together with the code. We do not use LLM during the process of constructing our benchmark.
>
> For (2), we fixed the random seed to be the default value 2024 provided within our submitted code.
>
> For (3), all prompt templates for evaluating LLMs are provided within Appendix G in the supplementary material.
>
> We also run every task for 800 examples, allowing us to observe that the metric stabilizes.

---

> ### Comment · Area_Chair_MMa9 · 2025-08-04
> **Please engage with authors during the discussion period**
>
> Dear reviewer wJaf,
>
> The authors have responded to your review with a rebuttal that addresses the four limitations that you noted. Please review the rebuttal that they have posted and respond to indicate whether your concerns have been addressed, or, if not, what information the authors should provide or what changes they should make to address them, to the degree that that is possible. There are only a couple of days left in the discussion period, and your engagement during this time is critical.

---

> ### Author Response · Authors · 2025-08-06
> **A Kind Reminder for Feedback**
>
> Dear reviewer wJaf,
>
> We have posted some experimental results (a new task in-context reasoning and results on the hallucination rate) to answer your concerns on 1) tasks being too simple, and 2) the trustworthy evaluation. We have also explained the misunderstanding about reproducibility. We would appreciate it if you could let us know whether our responses address your concerns.
>
> Look forward to your reply.
>
> Best regards,
>
> Authors

---

> ### Comment · Area_Chair_MMa9 · 2025-08-07
> **Please engage with the authors in a comment**
>
> Dear reviewer wJaf,
>
> Thank you for updating your 'final justification' in open review. The authors are unable to see that, though, and it will not be revealed to them until after decisions are out. Please engage with them by posting a comment so that they can see your response (it's fine if you want to just copy/paste your justification).

---

### Official Review · Reviewer_7FXB · 2025-07-02

**Rating:** 5
**Confidence:** 3

**Summary:**

This paper introduces LongBioBench, a benchmark for Long-Context Language Models (LCLMs) designed to address limitations in existing evaluation frameworks. The authors argue that current synthetic tasks often lack contextual coherence, while real-world tasks lack controllability. LongBioBench is built on three principles: seamless context (embedding coherent "needle" biographies within a "haystack" of other biographies), controllability (allowing systematic adjustment of task parameters), and sound evaluation (using objective metrics and preventing data contamination). The paper finds that Qwen and GPT-4o struggle with elementary reasoning and trustworthiness. The analysis also claims that long-context continual pre-training primarily boosts retrieval capabilities, with only marginal improvements to reasoning.

**Dataset Code Accessibility:**

Yes

**Dataset Code Comments:**

The dataset code is opensourced.

**Ethical Considerations:**

No, there are no or only very minor ethics concerns

**Final Justification:**

The authors have resolved all my major concerns. The authors adequately explained why hardness of openai MRCR provides for poor comparison, omitted their original claim on pretraining with long-context data and conducted more interesting correlation analysis with the HELMET dataset.

**Limitations Weaknesses:**

* Line 9-10: The first time reading these lines, while having seamless context alleviates the challenge, the need for controllable setting could be motivated better in writing. For e.g., what is being controlled can be mentioned in the abstract.
* Open AI MRCR [1] is another benchmark which offers context artificially generated to include distractors and needle in a seamless fashion and there is controllability in number of needles. It would be good to compare against it in Table 1 and perhaps get a correlation graph as in figure 1, but with mrcr. Also, comparing LongBioBench with openai mrcr would help the paper.
* The LongBioBench benchmark uses artificially generated biographies, a single and highly structured domain. While this allows for controlled analysis, the findings may not generalize to more diverse and unstructured real-world texts like legal documents or scientific papers. A model's success in parsing biographical data doesn't guarantee its performance on complex logical reasoning in other formats. Although the benchmark shows a strong correlation with the HELMET benchmark, its validity across a broader range of text types remains an open question.
* “Long context pretraining boosts retrieving, but does not help reasoning”: This claim is broad and out of scope of this paper:
    * How the 32k sequence length data are constructed is not mentioned, it could have disjoint documents with no relevant information across documents, which can explain why there is no improvement in reasoning.
    * The reasoning is measured with just 3 tasks in the middle graph in fig 7, which is too limited to make this claim. I encourage the authors to add more reasoning based evals (hence my argument that this claim is out of scope of this work).
    *  Additionally, the models such as Qwen could have pretraining data intersection with the data used for continual pretraining, which might lead to little change in reasoning performance. This claim should be made in a controlled setting.
* Helmet benchmark has correlation with the proposed dataset as mentioned in figure 1. Does the controllability of LongBioBench lead to a scenario where there is no correlation with helmet benchmark? it would be great to demonstrate this. This test could answer the question: Is longbiobench offering an additional signal that helmet can’t provide?

**Strengths Contributions:**

* The paper outlines an evaluation philosophy based on three criteria—seamless context, controllability, and soundness—that combines the benefits of synthetic generation with a more realistic design.
* The paper empirically demonstrates that incoherent context can lead models to use "shortcuts," using a "Bio-in-a-Haystack" experiment to show that models revert to identifying stylistically different text on harder tasks, which validates the benchmark's design.
* The study evaluates 18 models across multiple context lengths, includes an analysis of pre-training checkpoints to show how capabilities are acquired, and reports a correlation of r=0.853 with the HELMET benchmark to support its use as a proxy for applied tasks.
* The paper reports several findings, including that distractor density is a separate performance bottleneck and that some benchmarks may have a bias towards "numerical needles."

---

> ### Author Rebuttal · Authors · 2025-07-31
>
> Thank you for your review and the recognition of our seamless context construction, controlled task design and our findings. Please let us know if we have addressed your concerns.
>
>
> > Writing in the abstract
>
> Thank you for the suggestions. Here is the modified version of the abstract for L1-L10. Specifically, we add more explanations and an example for the controlled study.
>
> Existing frameworks for evaluating long-context language models (LCLM) can be broadly categorized into real-world applications (e.g, document summarization) and synthetic tasks (e.g, retrieve a synthetic information among unrelated long contexts--the needle-in-a-haystack (NIAH) task). Despite their utility, both approaches are accompanied by certain intrinsic limitations. Real-world tasks often involve complexity that makes interpretation challenging and suffer from data contamination, whereas synthetic tasks frequently lack meaningful coherence between the target information ("needle") and its surrounding context ("haystack"), undermining their validity as proxies for realistic applications. In response to these challenges, we posit that an ideal long-context evaluation framework should be characterized by three essential features: 1) coherent contextual integration between target information and its surrounding context; 2) an extensible task setup that enables controlled studies—*for example, incorporating additional required abilities such as numerical reasoning;* and 3) robust evaluation metrics, such as accuracy.
>
>
> > Comparison with OpenAI MRCR
>
> We agree that OpenAI MRCR is another useful synthetic benchmark with features similar to those of LongBioBench. Here is our comparison between MRCR and our LongBioBench.
>
> **MRCR is too hard to distinguish current open-sourced models, the results are too low for meaningful correlation calculation.**  We evaluated open-source models on MRCR using a 32k-token context window in a 2-needles setting (264 samples), but most models achieved an accuracy of under 10 percent—too low to distinguish among them or assess correlations with our benchmark meaningfully. The full results are presented in the following Table.  In contrast, LongBioBench is designed from the ground up with an incremental difficulty structure (e.g, the single retrieval could be easy for all models, but when it extends to calculation or multi-hop retrieval, some models fail). It enables clear differentiation across model capabilities: For instance, InternLM3-8b performs relatively well on retrieval tasks but struggles with reasoning scenarios.
>
> | Model                     | Accuracy |
> |--------------------------|----------|
> | Qwen2.5-7B-Instruct      | 0.2739   |
> | Qwen2.5-7B-Instruct-1M   | 0.2252   |
> | internlm3-8b-instruct    | 0.0992   |
> | glm-4-9b-chat-1m         | 0.0973   |
> | Llama-3.2-3B-Instruct    | 0.0671   |
> | Llama-3.1-8B-Instruct    | 0.0401   |
> | Llama-3.2-1B-Instruct    | 0.0401   |
>
> **Furthermore. It’s not straightforward to extend MRCR into other task types (e.g., numerical reasoning based on retrieved information.** Although MRCR also achieves a seamless insertion between the needle and haystack, the tasks in MRCR focus more on the pure retrieval of the correct paragraph (e.g., find the second poem) rather than reasoning over information within the context or measuring trustworthy output. Their fixed tasks (coreference resolution) make the benchmark unextensible to measuring different dimensions of abilities. We will add these comparisons to Table 1 in the camera-ready version
>
>
> > Validity across broader types of text
>
> **We test the generalization of our benchmark to the best of our knowledge.** We acknowledge that the synthetic biographies are not guaranteed to be generalized to the broad text in the real world. The generalization between the synthetic benchmark and the real-world benchmark has long been an open question [1] [2]. That’s why we perform the correlation test with the benchmark helmet, which covers the most comprehensive real-world context and tasks to the best of our knowledge.
>
> **We are more motivated to examine patterns from a controlled perspective.** In this work, we place a greater focus on conducting a controlled examination of the pattern of LCLMs. We believe that although successfully passing all the synthetic tests in the benchmark may not generalize to all tasks, the failing patterns can still provide insights and directions for optimization.
>
> [1] HELMET: How to Evaluate Long-Context Language Models Effectively and Thoroughly
>
> [2] NoLiMa: Long‑Context Evaluation Beyond Literal Matching
>
>
> > The claim on pretraining is out of scope
>
> We agree that the claim is too broad without decomposing the confounders of the pretrain data. We will delete this claim and add more observations. The modified version is:
>
> **No noticeable improvements on the reasoning abilities are gained through long-ciontext continual pretraining.** Comparing the middle figure with the left one, we observe that performance improves consistently across all retrieval tasks, whereas the reasoning task shows only a slight improvement, with accuracy remaining extremely low at around 10% (note that the random guess accuracy for the ranking task is 50% since it involves ranking two individuals). ~~This indicates that pretraining on longer contexts primarily enhances retrieval capabilities but not reasoning abilities~~. Interestingly, the calculation task follows a performance trajectory similar to all retrieval tasks and already achieves high accuracy before long-context pretraining. This suggests that Qwen2.5 already possesses the capability to perform calculations over relatively long contexts, and that the main bottleneck on this task lies in retrieval rather than reasoning. Therefore, we categorize the calculation task alongside the retrieval tasks.
>
> > Which current tasks are not correlated with Helmet dataset? Can we synthesize some tasks that are not correlated with that dataset?
>
> We thank the reviewer for providing this interesting angle. We perform correlation analysis on the sub-task and the results are as follows:
>
> | Task           | Spearman correlation | p-value |
> |----------------|----------|------------|
> | standard       | 0.92     | 0.00       |
> | multi_standard | 0.76     | 0.00       |
> | paraphrase     | 0.87     | 0.00       |
> | pronoun        | 0.86     | 0.00       |
> | rank           | **0.27**     | 0.39       |
> | calculation    | 0.83     | 0.00       |
> | twodiff        | **0.21**     | 0.50       |
> | multihop       | 0.78     | 0.00       |
> | citation       | 0.70     | 0.01       |
> | IDK            | 0.57     | 0.05       |
>
>
> **The two-diff task and rank task are the task that have least correlation with helmet.** This suggests that the two-diff tasks—which provide an open-ended setting with multiple correct answers and require constrained planning—may represent a new direction that Helmet struggles to evaluate. As for the rank tasks, the difficulty may lie in their simplicity; with random guessing yielding around 50% accuracy. That makes it challenging to differentiate model performance effectively.
>
> **Yes. We can sythesize such tasks use the controlability.** Two-diff task is therefore an example answer for this question.

---

> ### Comment · Area_Chair_MMa9 · 2025-08-04
> **Please engage with authors during the discussion period**
>
> Dear reviewer 7FXB,
>
> Thank you for posting the acknowledgment of the authors' rebuttal. Please note that the acknowledgment you added includes the phrase "I have engaged in discussions and responded to authors." An important part of the discussion period is the opportunity to discuss with the authors whether their rebuttal has addressed your concerns or, if not, what information the authors should provide or what changes they should make to address them, to the degree that that is possible. Please add a response to the rebuttal to indicate this. There are only a couple of days left in the discussion period, and your engagement during this time is critical.

---

> ### Author Response · Authors · 2025-08-06
> **A Kind Reminder for Feedback**
>
> Dear reviewer 7FXB,
>
> We have posted some experiment results on MRCR to answer your questions about the comparison with MRCR and included more correlation analysis. We have also updated our writing on the abstract and pretraining parts following your suggestions. We would appreciate it if you could let us know whether our responses address your concerns.
>
> Look forward to your reply.
>
> Best regards,
>
> Authors

---

### Official Review · Reviewer_1an3 · 2025-07-03

**Rating:** 5
**Confidence:** 4

**Summary:**

The authors argue that an ideal long-context evaluation framework does not exist, as the existing ones have either too much real-world complexity or are recall-based, with the to-be-recalled information being unrelated to context. They list essential features of an ideal benchmark. Equipped with these features, they present LongBioBench, a benchmark based on artificially generated biographies to evaluate long-context abilities across multiple dimensions. Their experiments show that most models lack semantic understanding and fail to perform simple reasoning over retrieved results, and are also less trustworthy with increased context length. The authors claim that their benchmark is better than existing alternatives (which they deem vulnerable to hacks) at measuring long-context abilities in a controlled setup while still being close to authentic language tasks.

**Additional Feedback:**

L\<NUM\> denotes a reference to line number \<NUM\>.

Some Corrections:
- L5: Should be "Needle-In-A-Haystack" instead of "Needle-In-The-Haystack" if abbreviated as NIAH
- L142: ...in our bench**mark**
- L257: You mention "The answer to this task is not determined", which is not true. Given the desired age difference and context, the answer is indeed determined. What you perhaps want to say is that the needles may vary over long contexts based on the age difference. Please rephrase accordingly.

**Dataset Code Accessibility:**

Yes

**Dataset Code Comments:**

I checked the benchmark on Kaggle, and looked at the code in the anonymous repository. The code appears to be well-documented and usable, thereby supporting reproducibility.

**Ethical Considerations:**

No, there are no or only very minor ethics concerns

**Final Justification:**

The premise of the paper is a long-context reasoning benchmark. While I liked the setup and design of the benchmark, I think it does not test "reasoning" as we know it. I differentiate between contextual pattern recognition (which is what the new ICL task is) and retrieval (which is what most of the tasks in the benchmark do) from reasoning (which means multiple steps of logical deduction or other reasoning variants). To that end, I find the paper sets an important premise but fails to fully deliver on it. Hence, I do not deem it to be technically flawless (which would attract a rating of 6). I still do believe this is an important work and needs to be published, hence my rating of 5.

**Limitations Weaknesses:**

- **Abstract.** I felt that the abstract does not do a good job of motivating the problem. While the problem description is there, it involves many undefined terms (seamless, sound evaluation, lack of coherence, etc.) used technically, and hence, the intended meaning is not coming out well. Also, what you mean by "real-world tasks" in abstract is unclear. An example like that of NIAH for synthetic tasks will be helpful. Please consider refining for readability.
- **Make the manuscript self-contained.** The concept of a distractor is used throughout the paper, but it is never defined. While it is intuitive and people familiar with related work can relate to it, I suggest including a definition. In general, please define all concepts before you use them. I also encourage the authors to include in the paper examples of specific questions used in the dataset.
- **While useful, benchmark still fails to meaningfully measure long-context reasoning.** I'd like to highlight some shortcomings regarding the tasks mentioned in section 2.3. Those tasks do not test long-context reasoning abilities effectively. It is only the retrieval (of the needle from the haystack) that is tested. This, to me, decreases the potential impact of this benchmark. With the setup put forth here (biographies), several interesting reasoning questions (albeit requiring manual effort) could be generated that relate multiple biographies across long contexts. I understand that "rank" and "twodiff" are good tasks, but they are primarily lookup-based. I do recognize the positive aspects of this benchmark and agree that it improves on the conventional NIAH benchmarks by introducing coherence. However, I also find it rather simplistic to be useful for highlighting fundamental long-context reasoning flaws in LCLMs.

**Strengths Contributions:**

- The paper highlights important limitations of existing long-context evaluation benchmarks across the dimensions of complexity (to the point that they fail to pinpoint specific limitations of models), lack of coherence between context and information to be recalled, and bias toward specific types of recalled information.
- I liked the list of features stated by authors that a long-context benchmark should have, especially the "seamless context" requirement, which makes the task close to a real-world setting and prevents it from being hacked.
- I found the paper to be well-written overall (see the next section for suggested improvements).

---

> ### Author Rebuttal · Authors · 2025-07-31
>
> Thank you for recognizing the value of our work, especially the seamless context features. Please let us know if we have addressed your concerns and questions.
>
>
> > Undefined terms in the abstract
>
> Thank you for the suggestions on the writing. Here is the modified version of the abstract for L1-L10. We added more explanation. The additional parts are marked in bold
>
> Existing frameworks for evaluating long-context language models (LCLMs) can be broadly categorized into real-world applications **(e.g., document summarization)** and synthetic tasks **(e.g., retrieving synthetic information among unrelated long contexts — the needle-in-a-haystack (NIAH) task)**. Despite their utility, both approaches are accompanied by certain intrinsic limitations. Real-world tasks often involve complexity that makes interpretation challenging and suffer from data contamination, whereas synthetic tasks frequently lack meaningful coherence between the target information ("needle") and its surrounding context ("haystack"), undermining their validity as proxies for realistic applications. In response to these challenges, we posit that an ideal long-context evaluation framework should be characterized by three essential features: **1) seamless context: coherent contextual integration between target information and its surrounding context; 2) controllable setting: an extensible task setup that enables controlled studies—for example, incorporating additional required abilities such as numerical reasoning; and 3) and sound evaluation: we avoid LLM-as-Judge and conduct exact-match to ensure our evaluation results are deterministic and highly reproducible.**
>
>
> > Definition of terms in the paper and question examples
>
> We will add the definition in the introduction and task examples in section 2.3.
>
> Similar to [1], our definition of distractors is the contextual contents that semantically state the same attributes. For example, “The hobby of Kennedy Quigley Shields is badminton” and “The hobby of Isabel Zeno Springer is satellite watching” are defined as mutually distractors in our benchmark. For examples of specific questions of tasks, we have illustrated them in Table 4 in the appendix. We will consider moving this table to our main context in the updated version. For instance, in the standard retrieval task, we have: “Context:”... The hobby of {P1} is dandyism….” Question: What’s the hobby of {P1}?”
>
> [1] NoLiMa: Long-Context Evaluation Beyond Literal Matching
>
>
> > The reasoning tasks are too simple to highlight fundamental long-context reasoning flaws.
>
> We have included “rank”, “calculation”, “multihop”, and “twodiff”, four types of tasks for testing numerical reasoning, multi-hop reasoning, and constraint planning abilities (illustrated in Figure 3). Our results demonstrated that the current settings are still challenging enough for the current LCLM. Moreover, our framework is designed to be extensible to more complex tasks. We added a new in-context learning (ICL) setting here as an example.
>
> **The tasks are still challenging for current LLMs.**
> We find that reasoning tasks remain challenging for most current open-source LLMs and even some closed-source models, such as GPT-4o-mini, which achieves only 39.6% average accuracy on reasoning tasks despite its strong retrieval performance (73% average accuracy). This highlights the difficulty LLMs face in jointly performing reasoning and retrieval (Results 1, L221). Among open-source models, the best-performing one—Qwen2.5-14B-1M—achieves just 58% average accuracy on reasoning tasks, with all other open-source models falling below 50%.
>
> **The extensibility of our framework makes it easy to construct more complex reasoning tasks.**
> We fully acknowledge that more creative and challenging reasoning tasks could help a lot. LongBioBench is highly extensible (L71) for more complex tasks since the context with fine-grained information is generated within a controlled framework. We introduce the following new ICL setting:
>
> Inspired by [2], which explores long in-context reasoning capabilities, we add a setting similar to the extreme label task:
> E.g. “{Bio1, Bio2}.
> Question: Which category of university did Charlotte Farley Hall graduate from?
> Answer: Category 3.
> …
> {Bio-n, Bio-n+1}
> Question: Which category of university did Gabriella Jenson Griffin graduate from?”
>
> In this setting, the context comprises multiple in-context demonstrations, each consisting of several biographies and a corresponding QA pair. For each QA pair, we construct a mapping between university names and categorical labels based on their initial letter (e.g., universities starting with “U” are assigned to Category 1, those starting with “A” to Category 3, etc.). Importantly, this mapping is not explicitly provided to the model. Instead, LCLMs are expected to infer the underlying rule by generalizing from the in-context examples. To prevent LCLMs from exploiting memorization or retrieval of the same university name, we ensure that the queried university is unique. However, to support reasoning, we guarantee that at least one university in the demonstrations shares the same initial character, providing a subtle hint for the mapping.
>
> We perform experiments over 10 uniformly distributed categories on the best model Qwen2.5-7b-Instruct-1M and Qwen2.5-14b-Instruct-1M with a cot prompt. The results are as follows:
>
> | ICL           | 2k   | 8k    | 16k   | 32k   | 64k   | 128k  |
> |---------------|------|------|------|------|------|------|
> | Qwen2.5-14B-Ins-1M | 51.5 | 25.5 | 30.5 | 25   | 16   | 17   |
> | Qwen2.5-7B-Ins-1M | 20   | 19.5 | 11   | 11.5 | 10.5 | 12.5 |
> |Random Guess|10|10|10|10|10|10|
>
> We observe the following interesting findings:
> - As the context length increases, we provide more demonstrations for LCLMs to learn from. However, we still observe a significant performance drop from 8k -> 16k for Qwen7b and 2k -> 8k for Qwen14b. This suggests that the in-context learning abilities can still be negatively impacted when LCLMs handle longer contexts.
> - Even current SOTA LCLMs still struggle with in-context learning over long inputs
> - Observing the failure cases, we found an interesting pattern where the model hallucinates the category mapping it found through their thinking process, even though they have strong retrieval abilities, which could be a promising optimization direction for current LCLMs (especially for those thinking models)
>
> [2] LongICLBench: Long-context LLMs Struggle with Long In-context Learning
>
>
> > Some corrections
>
> We thank the reviewer for highlighting these corrections. We will incorporate all suggested changes.

---

> > ### Comment · Reviewer_1an3 · 2025-08-02
> > **Thanks for the rebuttal**
> >
> > I thank the authors for their rebuttal. My evaluation of this work continues to be positive, and I maintain my rating of 5.

---

### Author Response · Authors · 2025-08-05
**General Comment**

We thank all the reviewers for providing thoughtful reviews. We are glad that reviewers generally recognize our strengths/contributions:

- Our work demonstrates the importance of coherent context in long-context evaluation (**1an3**, **7FXB**, **wJaf**, **jv1z**), as incoherent context may lead models to utilize shortcuts for answering long-context questions (**1an3**, **7FXB**).
- Our proposed LongBioBench features coherent needle insertion (Seamlessness), controlled evaluation (Controllability), and sound metrics with contamination-free evaluation (Soundness) (**1an3**, **7FXB**, **wJaf**, **jv1z**).
- LongBioBench has a high correlation (0.853) with the comprehensive non-synthetic benchmark Helmet (**7FXB**, **jv1z**), thus still being close to authentic tasks (**1an3**)
- Through controlled evaluation, our work provides interesting insights (**wJaf**), including (1) distractor density is another main bottleneck for long-context retrieving (**7FXB**); (2) models are biased on specific types of needles (**1an3**), such as number (**7FXB**); (3) reasoning is still a main pain-point, even though models are capable of retrieving (**1an3**); and (4) the performance trend on long-context pretraining (**jv1z**)

Furthermore, we want to highlight the **extensibility** of LongBioBench (illustrated in Table 1 and Figure 2), which enables flexible yet controlled increases in task complexity. Guided by the cognitive capabilities we want to examine, our tasks are created in a progressively hierarchical manner. For example, to assess retrieval $\rightarrow$ numerical reasoning $\rightarrow$ constraint planning abilities, we design standard (retrieve a specific attribute of one bio) $\rightarrow$ calculation (return the age difference of people) $\rightarrow$ Two-diff (return two people whose age difference is a given number) tasks correspondingly. This framework not only allows for controlled comparisons across tasks using the same context, but more importantly, provides a scalable framework to flexibly incorporate new tasks based on the abilities we seek to measure.

We try to address the reviewer's common concern as follows:

> **Common concern: Motivation in the abstract should be clearer (1an3,  7FXB)**

We have provided an updated version in the responses, which includes more explanations of technical terms, additional examples, and justifications.

> **Common concern: Tasks within the longbiobench are relatively simple (1an3, wJaf)**

As mentioned above for extensibility, we had several simple settings for measuring specific fundamental capabilities of LLMs. We stopped proposing more complex problems when we found that most long-context models struggle with the proposed tasks (SOTA models, such as Qwen2.5-14B-1M, only achieve 58% on average in our proposed reasoning tasks).  Nonetheless, to demonstrate the extensibility of our benchmark, we have introduced an additional setting—*In-context Reasoning*—as an example of a more challenging task.

Rather than focusing on simple numerical reasoning or retrieval, the task requires the model to reason across all provided demonstrations in the long context (a group of bios and their labels). Specifically, the model must infer an implicit mapping between university names and their corresponding categories based on the first character, using only the provided demonstrations as guidance. We perform experiments on the best model, Qwen2.5-7b-Instruct-1M and Qwen2.5-14b-Instruct-1M, with a chain-of-thought prompt, and have the following results:

| ICL (Acc)        | 2    | 8    | 16   | 32   | 64   | 128  |
|---------------|------|------|------|------|------|------|
| Qwen2.5-14B-Ins-1M | 51.5 | 25.5 | 30.5 | 25   | 16   | 17   |
| Qwen2.5-7B-Ins-1M | 20   | 19.5 | 11   | 11.5 | 10.5 | 12.5 |
|Random Guess|10|10|10|10|10|10|

We observe the following findings:
- Even current SOTA LCLMs still struggle with in-context reasoning over long inputs.
- As the context length increases, we provide more demonstrations for LCLMs to learn from. However, we still observe a significant performance drop from 8k -> 16k for Qwen7b and 2k -> 8k for Qwen14b, suggesting that the in-context learning ability is limited by the model’s ability to handle long context.
- Observing the failure cases, we found that though the model demonstrates strong retrieval ability at the given context, it again begins to hallucinate within their reasoning process, indicating a gap between being able to retrieve and to further reason about the retrieved information. This could be a promising research direction for current LCLMs (especially for those long-cot models).

We will also include these supplementary results and discussions in our final version.

We once again thank the reviewers for their insightful comments, and don't hesitate to contact us if there is anything else we can do to help you better understand and recommend our paper.

---

### Decision · Program_Chairs · 2025-09-18

**Decision:**

Accept (spotlight)

**Comment:**

**Summary**: This paper proposes LongBioBench, a benchmark that addresses flaws in existing long-context evaluations by using synthetically generated biographies. This approach creates controllable tasks to assess a model's understanding, reasoning, and trustworthiness. They find that current models perform poorly on reasoning as context length increases.

**Strengths**:
- Emphasis on coherent/seamless context is an improvement over tasks that rely on “needle in a haystack” setups, which also makes the task more closely approach a realistic use case (1an3, 7FXB)
- Controllable and extensible framework enables precise studies with detailed analysis (7FXB, jv1z)
- Findings are novel and of interest to multiple communities (7FXB, jv1z)
- Overall paper is well-written (1an3, wJaf)

**Weaknesses**:
- The benchmark only tests retrieval, not long context _reasoning_ (1an3), and the tasks may be too simple (wJaf). The authors reply in the rebuttal that the framework is flexible and more complex tasks that test reasoning could follow their framework in the future, but that still does not address the point that the benchmark does not effectively probe model reasoning ability. The authors quickly propose a potential task in the rebuttal, but this is a much larger portion of the research that can’t be fairly evaluated on the short timeframe of the discussion period.
- Lack of clear definitions for terms before they’re used makes the paper less readable (1an3, 7FXB). The authors commit to addressing this in future drafts.
- Missing comparison to a similar benchmark (7FXB). The authors provide a comparison in the rebuttal, but their main differentiation seems to be that scores on the similar benchmark are lower than in LongBioBench, but that may actually work against the authors, as it seems to indicate that the other benchmark is more difficult and thus likely to have more staying power. However, the authors also point out that their benchmark can be more easily extended.
- Benchmark may have poor generalizability due to it being synthetic (7FXB, wJaf). Authors argue the high correlation with the HELMET benchmark mitigates this concern.
- Initial claims about the effects of long-context pretraining were too broad (7FXB). Authors retracted the original claim in the rebuttal and replaced it with a more nuanced observation, which resolved the issue for the reviewer.

**Overview / explanation of recommendation**: This paper proposes a controllable benchmark with a more realistic setup than previous work, and is a clear improvement on standard "needle-in-a-haystack" evaluations for long context models. Reviewers agree the paper is well-motivated, well-designed, and yields interesting insights. The primary weakness of the work is that the benchmark, as presented, fails to meaningfully evaluate long-context _reasoning_. The framework being extensible means that this concern is potentially resolvable (and the authors propose another task in the rebuttal), but with respect to the current paper I don’t find this concern to be fully resolved, though the benchmark is still a well-executed and meaningful contribution.

===== FINAL UPDATE FROM DB Track PCs ====

The final decision for this paper has been taken by the program chairs after consultation with the SACs. All Senior Area Chairs have ranked papers according to the feedback from the AC during the review process. We decided to leave the original meta-review to reflect the opinion of the AC in light of the initial discussions with reviewers and SAC.